# Sample-Efficient Multi-Round Generative Data Augmentation for Long-Tail Instance Segmentation

**Byunghyun Kim, Minyoung Bae, Jae-Gil Lee***
KAIST
{rooknpown, mybae, jaegil}@kaist.ac.kr

## Abstract

Data synthesis has become increasingly crucial for long-tail instance segmentation tasks to mitigate class imbalance and high annotation costs. Previous methods have primarily prioritized the selection of data from a pre-generated image object pool, which frequently leads to the inefficient utilization of generated data. To address this inefficiency, we propose a *collaborative* approach that incorporates feedback from an instance segmentation model to guide the augmentation process. Specifically, the diffusion model uses feedback to generate objects that exhibit high uncertainty. The number and size of synthesized objects for each class are dynamically adjusted based on the model state to improve learning in underrepresented classes. This augmentation process is further strengthened by running *multiple rounds*, allowing feedback to be refined throughout training. In summary, *multi-round collaborative augmentation (MRCA)* enhances sample efficiency by providing optimal synthetic data at the right moment. Our framework requires *only* 6% of the data generation needed by state-of-the-art methods while outperforming them.

## 1 Introduction

The advent of diffusion models has made generating high-quality images significantly easier with lower cost[9, 23, 30, 31]. These advancements naturally opened new possibilities for generating training data in data-hungry vision tasks like object detection[16, 17] and instance segmentation[4, 21, 27][1]. Diffusion models can synthesize images conditioned on textual descriptions, enabling the creation of diverse and detailed datasets with labels[6, 22]. Especially for long-tail instance segmentation, where class imbalance and high annotation costs pose significant challenges, diffusion-based image synthesis has become a crucial solution.

Copy-paste-based methods, originally intended for augmenting training data using *real* image objects[8, 15], now have been extended to utilize high-quality *synthetic* image objects generated by diffusion models. That is, these recent methods first generate a large pool of image objects and then augment the training data by pasting them to real images[11, 37, 41]. The advantage arises from generating objects in a *training-free* manner, at a comparably lower cost than layout-based methods that train the diffusion model with the training data as shown in Figure 1a[5, 40]. However, some studies have shown that continuously training with synthetic data can eventually degrade model performance[12, 13]. Such finding has naturally led to research on active and curriculum learning strategies that can incrementally select most informative synthetic objects from a pre-generated object pool as shown in Figure 1b[26, 41]. Nevertheless, *preemptively* constructing the object pool before training may be suboptimal, as it can result in a significant number of unused objects and may not include object characteristics that could address the model's deficiencies encountered during training.

---

*Corresponding author

[1]While we designate instance segmentation as our target task, object detection is equally relevant.

39th Conference on Neural Information Processing Systems (NeurIPS 2025).

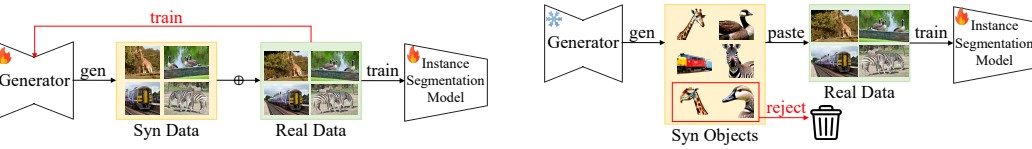

(a) Layout-based generative augmentation (existing).    (b) Training-free generative augmentation (existing).

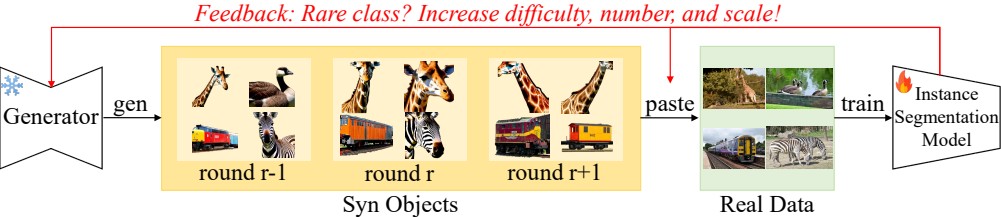

(c) MRCA: multi-round collaborative augmentation (ours).

Figure 1: *Novelty* of our generative augmentation strategy. Existing methods **(a)** train a generator separately or **(b)** discard part of pre-generated data and incur extra cost. Ours **(c)** incorporates feedback from the training model into the generation process, optimizing cost and enhancing quality.

To solve this problem, we propose *multi-round collaborative augmentation* between a generator and an instance segmentation model to guide the generator to build an augmented training set which can achieve high accuracy with a smaller number of synthetic objects—i.e., toward being *sample-efficient*.

**Collaborative Augmentation**    To effectively improve the power of the instance segmentation model using synthetic images, we transfer its feedback to the augmentation process. As illustrated in Figure 1c, the feedback is exploited in *two* distinct places: (i) in the generation of synthetic objects and (ii) in the pasting of these objects to real images. For the generation step, the classifier gradients are employed to produce more challenging objects for the instance segmentation model's classifier, akin to prior research on image classification [1, 19]. Also, the class-wise validation accuracy and classifier weights are used to ascertain an *adequately mandated quantity* of synthetic objects for each class, thereby enhancing the classifier's capacity to learn from underrepresented or difficult classes. For the pasting step, the class-wise validation accuracy is employed to assign greater space to underperforming classes, as larger objects are more effective in enhancing the performance of instance segmentation [28]. In short, this *two-fold* feedback approach facilitates the object generator in supplying the most beneficial data to the instance segmentation model.

**Multi-Round**    This novel collaborative augmentation is boosted *through multiple rounds*. It is well-known that the most useful training instances vary depending on the training stage, as witnessed by curriculum learning [3]. Thanks to the collaborative augmentation, the feedback from the instance segmentation model can be effectively transferred to the object generator. Then, it is more effective to transfer such feedback periodically throughout the training stage. As a result, the generation and pasting of synthetic objects occur *incrementally*, resulting in a reduced overall quantity of synthetic objects. Regarding each incremental generation, better customized images can be supplied for the training of the instance segmentation model at the appropriate time.

In summary, our proposed **M**ulti-**R**ound **C**ollaborative **A**ugmentation (**MRCA**) achieves higher or comparable performance with *only* 6% of object generation compared to state-of-the-art methods [7, 37, 41] in generative augmentation for long-tail instance segmentation on the LVIS 1.0 [18] dataset, particularly outperforming in rare classes. Extensive ablation experiments show the effectiveness of the two-fold feedback components of MRCA. The source code is publicly available at https://github.com/kaist-dmlab/MRCA.

## 2    Related Work

**Data Synthesis for Instance Segmentation**    Data synthesis for instance segmentation can be broadly categorized into two main approaches. The former involves cut-and-paste (or copy-paste-based) methods, where objects from a source image are extracted with precise boundaries and pasted onto destination images to create new synthesized images. This approach originates from cut-and-paste augmentation [8] and copy-paste augmentation [15], which were initially designed for pasting real objects to enhance data diversity.

Building upon these foundational methods for the former approach, X-Paste[37] extends the basic cut-and-paste method by incorporating objects from multiple sources, including both real and diffusion-based synthetic objects, to further diversify training examples. More recently, BSGAL[41] builds upon X-Paste by integrating instance-wise data selection strategies based on gradient and loss information, ensuring that pasted objects contribute effectively to model training. Additionally, DiverGen[11] focuses on diversity-centered instance generation, optimizing synthetic data to enhance representation learning by promoting diversity among generated objects.

The latter approach involves layout-based methods, where bounding boxes or segmentation masks are given a priori for generating images for object detection. GeoDiffusion[5] and ODGEN[40] generate bounding box coordinates derived from the dataset and condition the diffusion model to produce objects accordingly. InstaGen[14] instead synthesizes images and creates bounding box annotation by training an instance-level grounding head. However, these methods cannot be extended to instance segmentation tasks because of the high complexity of segmentation masking. Even for layout-based methods that provide segmentation masks as a condition[25], the diverse geometric knowledge of the diffusion model cannot be fully utilized owing to the fixed segmentation mask.

**Diffusion with Guidance**     The diffusion model generates the data following a data distribution $p(x)$ by learning the denoising process. Recent diffusion models are based on classifier-guidance[6] or classifier-free guidance[22] that learns the data distribution conditioned on class labels $p(x|y)$. That is, $\nabla_x \log p_{\theta,\gamma}(x|y) = \nabla_x \log p_\theta(x) + \gamma \nabla_x \log p(y|x)$, where $p_\theta(x)$ is the generative model's data distribution of an image $x$ parametrized by $\theta$, $p(y|x)$ is the classifier's probability of a class $y$ given $x$, and $\gamma$ is a guidance weight controlling classifier influence.

ControlNet[36] proposed a more structured approach by injecting additional spatial conditions into the diffusion process without retraining the base model, allowing fine-grained control over outputs. More recently, Universal Guidance[2] emerged as a general framework that can direct diffusion models using arbitrary signals through plug-and-play mechanisms, broadening the flexibility and applicability of guided generation. Altogether, these methods have significantly expanded the controllability and reliability of diffusion-based generative models.

## 3   Methodology: MRCA

### 3.1   Problem Formulation

An instance segmentation model is trained using a training set $\mathcal{D}^{real} = \{(x_i, \mathbf{y}_i)\}|_{i=1}^T$, where $x_i$ is an image, $\mathbf{y}_i$ is a set of annotations for the image $x_i$, and $T$ is the total number of images. Here, $\mathbf{y}_i = \{y_{i,j}|y_{i,j} = (bb_{i,j}, m_{i,j}, c_{i,j})\}|_{j=1}^{|\mathbf{y}_i|}$, where $bb_{i,j}$, $m_{i,j}$ and $c_{i,j}$ denote the bounding box, segmentation mask, and class of the $j$-th object.

We aim to augment the original training set $\mathcal{D}^{real}$ through *data synthesis* to increase data diversity, rectify class imbalance, and enhance model generalization. To achieve this goal, a set of image objects $\mathcal{O}^{syn} = \{o_i \mid o_i = (I_i, y_i)\}|_{i=1}^B$ is created using a generative model $\mathcal{G}$ with a given label $y_i$ from the set of classes as well as a dichotomous segmentation model $\mathcal{S}$, where $B$ represents the given budget of data synthesis. Two functions[2] are internally used for this augmentation process $\mathcal{A}(\mathcal{D}^{real}) = \mathcal{D}^{aug}$, as shown in Figure 2.

- Gen($B$): It generates $B$ image objects to compose $\mathcal{O}^{syn}$, where the number of objects per class is automatically determined by the learning progress of the instance segmentation model.
- Map-Paste($o$): For each object $o \in \mathcal{O}^{syn}$, it first selects the real image and annotations, $(x, \mathbf{y}) \in \mathcal{D}^{real}$, that $o$ will be pasted. Consequently, the mapping between objects and images, $\mathcal{P}$, is constructed. For each pair of an object and an image $(o, (x, \mathbf{y})) \in \mathcal{P}$, it then pastes $o$ into $x$ and accordingly updates the corresponding set of annotations $\mathbf{y}$. That is, it returns an augmentation $(x', \mathbf{y}')$ from $(x, \mathbf{y}) \in \mathcal{D}^{real}$ and $o \in \mathcal{O}^{syn}$.

As a result, an augmented training set $\mathcal{D}^{aug} = \{(x'_i, \mathbf{y}'_i)\} |_{i=1}^T$ is used instead of $\mathcal{D}^{real}$. Note that the number of images is fixed to $N$ whereas the number of annotations in these images is increased by $B$, following the common convention[11, 37, 41]. It is also possible to expand the set of images that are used for augmentation, and we leave it as a topic of future work.

---

[2]Auxiliary arguments (e.g., a generative model) of the functions are omitted for simplicity.

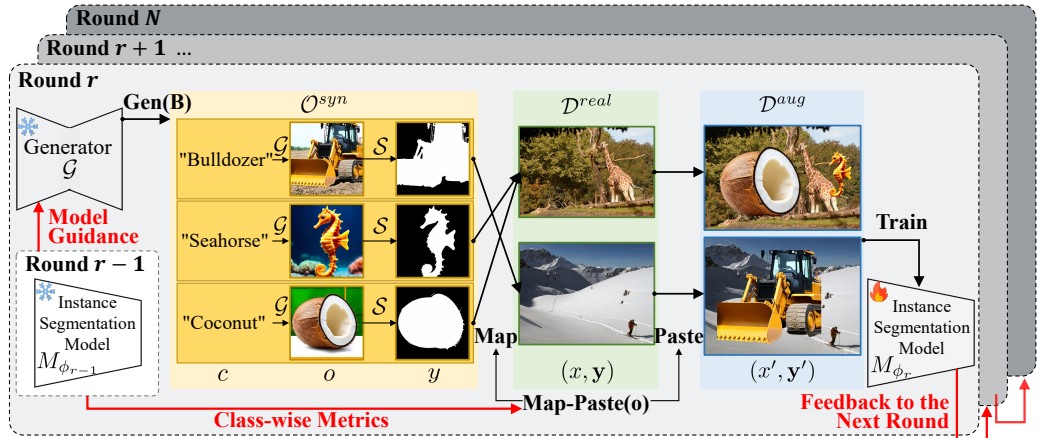

Figure 2: Inside of the multi-round collaborative augmentation with Gen(·) and Map-Paste(·). The model guidance and class-wise metrics are delivered from the model of the previous round to the copy-paste pipeline in the current round to optimize the Gen(·) and Map-Paste(·) processes.

Let's consider two instance segmentation models trained on $\mathcal{D}^{real}$ and $\mathcal{D}^{aug}$, respectively. The performance difference between these models on the test set $\mathcal{D}^{test}$ indicates the quality($Q$) of the data augmentation. Therefore, the primary challenge is to find an augmentation process $\mathcal{A}_B$ that *maximizes the quality of data augmentation within a specified budget $B$ with respect to $\mathcal{D}^{real}$*. Specifically, let $M^{real}$ and $M^{aug}$ denote the instance segmentation models trained on $\mathcal{D}^{real}$ and $\mathcal{D}^{aug}$, respectively, and $\mathcal{L}$ denote the loss function. Then, the objective is formalized as finding $\mathcal{A}_B$ such that

$$\arg\max_{\mathcal{A}_B} Q(\mathcal{A}_B(\mathcal{D}^{real})) = \arg\max_{\mathcal{A}_B}\{-\mathbb{E}_{(x,\mathbf{y})\sim\mathcal{D}^{test}}\mathcal{L}(M^{aug}(x),\mathbf{y}) + \mathbb{E}_{(x,\mathbf{y})\sim\mathcal{D}^{test}}\mathcal{L}(M^{real}(x),\mathbf{y})\}.$$

(1)

## 3.2 Multi-Round Collaborative Augmentation

To address the challenge, we propose the framework of *multi-round collaborative augmentation* (MRCA), which exploits the *feedback* from an instance segmentation model *through multiple rounds*. Figure 3 shows its overall procedure. MRCA consists of a warm-up stage and $N$ rounds, where each round comprises one or several epochs of the instance segmentation model training. We initially run the warm-up stage by training the instance segmentation model with real data and generating objects with no feedback. After the warm-up stage or a round is completed, the feedback from the instance seg-

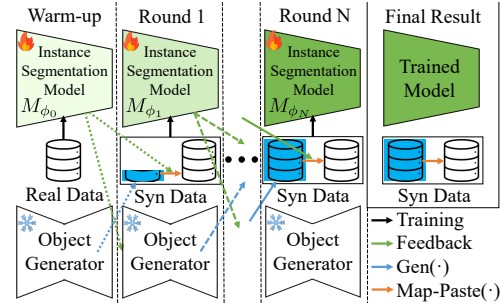

Figure 3: Multi-round collaborative augmentation.

mentation model is applied to both Gen(·) and Map-Paste(·) of the next round. Therefore, the instance segmentation model and the object generator *collaboratively* augment the training data. See Appendix A.1 for the pseudocode of this augmentation pipeline.

For every round, the training of the instance segmentation model and the generation of image objects are done simultaneously on multiple GPUs. However, the training is dependent on the generation since the generated objects are required for the augmentation of the training set. To overcome the delay caused by the dependency, the generated objects are passed to the next round for training. In summary, at the round $r$, we generate objects with the feedback from the previous round $r-1$ and provide the generated objects to the next round $r+1$.

**Budget Constraint**   We set the generation budget $B$ equal to the maximum number of generations possible in the training time of each round. By doing so, both the generation and training steps can overlap on different GPUs and thus are processed without delays.

**Collaborative Augmentation**   Given the budget constraint, our main idea is to leverage feedback from the instance segmentation model $M$ during the augmentation process. The intuition is that the

model $M$ holds valuable information that can help generate the most useful objects in Gen($\cdot$) and guide optimal placement during Map-Paste($\cdot$), thereby using the budget more effectively. Employing model feedback for generative data augmentation has been shown effective in classification tasks by directly aligning augmentation with the model's objective as in previous studies [1, 34]. We formalize this widely-accepted idea as Assumption 3.1 with specific conditions.

**Assumption 3.1** (Collaborative Augmentation Improves Quality). *Let $\mathcal{A}$ denote a data augmentation applied to a dataset $\mathcal{D}$, and let $f_\phi$ denote the feedback from a model $M_\phi$ trained on the dataset $\mathcal{D}$. Then, under the conditions of (C1) sufficient model capacity, (C2) controlled distribution shift, and (C3) proper feedback direction, the augmentation guided by $f_\phi$ has higher quality than the unguided augmentation. That is,*

$$(C1) \wedge (C2) \wedge (C3) \rightarrow Q(\mathcal{A}(\mathcal{D}; f_\phi)) \geq Q(\mathcal{A}(\mathcal{D})). \tag{2}$$

**Conditions for the Assumption**    We first define the information metric $\mathcal{I}_\mathcal{D}(M_\phi)$ as a measure of how well the model $M_\phi$ captures the distribution of the dataset $\mathcal{D}$. It can be measured by the average accuracy, i.e., $\mathcal{I}_\mathcal{D}(M_\phi) = \frac{1}{|\mathcal{D}|} \sum_{(x,\mathbf{y}) \in \mathcal{D}} \frac{1}{|\mathbf{y}|} \sum_{y \in \mathbf{y}} \mathbf{1} \left[ \exists \hat{y} \in M_\phi(x) \text{ s.t. IoU}(y, \hat{y}) \geq 0.5 \right]$.

- (C1) Sufficient model capacity: Model capacity can be represented by the empirical risk $\mathcal{R}_{emp}$, which is defined as the average loss $\mathcal{L}$ over the dataset $\mathcal{D}$. Formally, $\mathcal{R}_{emp}(M_\phi) = \frac{1}{|\mathcal{D}|} \sum_{(x_i, \mathbf{y}_i) \in \mathcal{D}} \mathcal{L}(M_\phi(x_i), \mathbf{y}_i)$. To ensure that the model retains available learning capacity, the empirical risk should be decreasing over time, i.e., $\frac{d\mathcal{R}_{emp}(M_\phi^{(t)})}{dt} < 0$.
- (C2) Controlled distribution shift: The knowledge gained through augmentation should surpass the adverse effect of distribution shift caused by the augmentation. Formally, given the data distribution $p_\mathcal{D}$ and the model's learned distribution $p_{M_\phi}^{(t)}$ over time $t$, $\frac{dD_{KL}(p_\mathcal{D} || p_{M_\phi}^{(t)})}{dt} < 0$.
- (C3) Proper feedback direction: Given the information from the model $M_\phi$ on the dataset $\mathcal{D}$, the feedback $f_\phi$ should properly guide the augmentation process toward quality maximization. For example, the feedback should be applied in the direction that increases diversity and uncertainty of the dataset (proper direction), not decreasing them (improper direction). That is, $[\mathcal{I}_\mathcal{D}(M_\phi) \geq \mathcal{I}_\mathcal{D}(\emptyset)] \wedge (C3) \rightarrow Q(\mathcal{A}(\mathcal{D}; f_\phi)) \geq Q(\mathcal{A}(\mathcal{D}))$.

Under the sufficient model capacity and controlled distribution shift conditions, the average accuracy of the model $M_\phi$ on the dataset $\mathcal{D}$ increases through training according to the definitions. That is, $(C1) \wedge (C2) \rightarrow \mathcal{I}_\mathcal{D}(M_\phi) \geq \mathcal{I}_\mathcal{D}(\emptyset)$. Combining it with the proper feedback direction condition $(C3)$, we obtain the overall guarantee that $Q(\mathcal{A}(\mathcal{D}; f_\phi)) \geq Q(\mathcal{A}(\mathcal{D}))$ holds under these conditions.

Similarly, as the model is trained by the data with higher quality of augmentation, the quality of the feedback from the model will increase accordingly. We incrementally formalize this concept as Lemma 3.2.

**Lemma 3.2** (Collaborative Augmentation with a Further Trained Model Improves Quality). *Let $\mathcal{A}$ be a data augmentation applied to a dataset $\mathcal{D}$, and let $f_{\phi_r}$ be the model feedback obtained from a model trained for $r$ rounds. Then, under Assumption 3.1 with the three conditions ($C1$, $C2$, and $C3$), the augmentation guided by a further trained model $f_{\phi_r}$ provides higher quality than the one guided by $f_{\phi_{r-1}}$. That is,*

$$Q(\mathcal{A}(\mathcal{D}; f_{\phi_r})) \geq Q(\mathcal{A}(\mathcal{D}; f_{\phi_{r-1}})), \quad \text{for } r > 0. \tag{3}$$

*Proof.* By Assumption 3.1, under sufficient model capacity ($C1$) and controlled distribution shift ($C2$), $\mathcal{I}_\mathcal{D}(M_{\phi_r}) \geq \mathcal{I}_\mathcal{D}(M_{\phi_{r-1}})$. Also, more information guides augmentation of higher quality under proper feedback direction ($C3$), so $(C1) \wedge (C2) \wedge (C3) \rightarrow Q(\mathcal{A}(\mathcal{D}; f_{\phi_r})) \geq Q(\mathcal{A}(\mathcal{D}; f_{\phi_{r-1}}))$. $\square$

**Multi-Round Augmentation**    We further claim that applying feedback in multiple rounds is superior to a one-time feedback under certain conditions. The high-level idea is that iterative augmentation incorporates feedback from a recently trained model instance, ensuring that the augmented data aligns with the model's evolving needs. We formalize this statement in Theorem 3.3.

**Theorem 3.3** (Multi-Round Collaborative Augmentation Improves Quality). *Let $\mathcal{A}$ be a data augmentation applied to a real dataset $\mathcal{D}_0 = \mathcal{D}^{real}$, and let $f_{\phi_r}$ denote the model feedback obtained*

*from a model $M_{\phi_r}$ trained on the dataset available at the round $r$. Consider a multi-round augmentation, where augmentation is performed over $N$ rounds. The augmented dataset at the round $r$ with cumulative budget $\frac{r}{N}B$ is defined as*

$$\mathcal{D}_{r,\frac{r}{N}B} = \mathcal{A}_{\frac{1}{N}B}(\mathcal{D}_0; f_{\phi_{r-1}}, \cup_{i=1}^{r-1}\mathcal{O}_i^{syn}), \quad for\ r = 1, 2, \cdots, N, \quad |\mathcal{O}_i^{syn}| = \frac{1}{N}B. \quad (4)$$

*Then, multi-round collaborative augmentation has higher quality compared to a single-round collaborative augmentation. That is,*

$$Q(\mathcal{D}_{N,B}) \geq Q(\mathcal{D}_{1,B}), \quad for\ N > 1. \quad (5)$$

*Proof.* The detailed proof is provided in Appendix B. □

To ensure that the theoretical claims hold in practice, we specifically design MRCA to keep the assumption better. First, we run the training process until the model validation accuracy saturates to verify that the current round has more information than the previous round, $\mathcal{I}_{\mathcal{D}}(M_{\phi_r}) \geq \mathcal{I}_{\mathcal{D}}(M_{\phi_{r-1}})$. While this statement is usually true since the model learns from newly generated objects, learning on only a portion of the dataset sometimes leads to biases and information loss. Therefore, we train the model for multiple epochs between rounds to incorporate the entire dataset. Second, the feedback $f_{\phi_r}$ itself should be properly applied to the augmentation process toward the direction that maximizes the quality of augmentation. Since the mechanism of how feedback improves augmentation quality solely depends on its design, we explain the details of how each feedback guides the augmentation process toward quality maximization in the next section.

### 3.3 Feedback on Object Generation

**Class-Wise Budget Optimization**    Class-wise budget optimization dynamically adjusts the number of synthetic objects for each class based on the model's state. Specifically, the number of objects to create for a class $c \in \{1, \cdots, c_{max}\}$ at the round $r$, $B_r(c)$, is computed as

$$r \leq 2 : B_r(c) = \frac{B}{c_{max}}, \quad r > 2 : B_r(c) = \alpha_r(1 - A_{r-2,c}) \cdot \|C_{r-2,c} - C_{r-3,c}\| \cdot \frac{1}{S_{r-2,c}}, \quad (6)$$

where $C_{r,c}$ is the classifier weight, $A_{r,c}$ is the validation accuracy, and $S_{r,c}$ is the number of objects in the augmented training set. The scale factor $\alpha_r$ scales the budget for each round so that the sum of $B_r(c)$ is equal to $B$. Eq. (6) targets generation on classes with low accuracy, low coverage, and scarce data. It is straightforward how budget is weighed on classes with low validation accuracy and a low number of training instances for sample efficiency. The classifier weight $C_{r,c}$ is defined as the weight vector corresponding to a class $c$ in the final classification weight matrix at the round $r$. The classifier weight difference term $\|C_{r-2,c} - C_{r-3,c}\|$ reflects how much the class has learned between the rounds, indicating room for improvement. Therefore, we provide more budget to classes with larger weight differences to maximize the learning efficiency.

The two-round gap between the left-hand term and the right-hand term in Eq. (6) results from two factors. One round gap occurs because the generation uses metrics from the model trained on the previous round. Another round gap is required to earn time for generating objects. This intentional gap allows efficient pipelining of generation with feedback, resulting in minimal halts.

**Feedback-Guided Object Generation**    To imbue each object with richer class information, we incorporate feedback from the instance segmentation model $M_\phi$ into the diffusion process. Specifically, the diffusion sampling is steered by the gradients of $M_\phi$ with respect to our chosen criterion $C$. By using entropy as the criterion, we synthesize objects that become progressively more difficult for the model to learn over successive rounds. Mathematically, the diffusion process is conditioned as

$$\nabla_x \log p_{\theta,\gamma,\omega}(x|y) = \nabla_x \log p_\theta(x) + \gamma\nabla_x \log p(y|x) + \omega\nabla_x \mathcal{C}(x, y, M_\phi), \quad (7)$$

where $\gamma$ is the scale of classifier guidance and $\omega$ is the scale of feedback guidance. Compared to previous work [1] on a classifier model, computing feedback guidance from an instance segmentation model requires an additional technique. A general instance segmentation model produces multiple objects' bounding boxes and segmentation masks from a given image which cannot be directly used for guidance. Fortunately, we aim to generate a single object per image in our method. Therefore, we set the whole image as the box proposal and compute the remaining layers of the instance segmentation model. The detailed algorithm is provided in Appendix A.2.

### 3.4 Feedback on Map-Paste of Objects

Recall the two stages of the Map-Paste process defined in Section 3.1. The first stage is *Map*, where we create a set of mappings $\mathcal{P}$ between $o \in \mathcal{O}^{syn}$ and $(x, \mathbf{y}) \in \mathcal{D}^{real}$. The second stage is *Paste*, where each mapped object in $(o, (x, \mathbf{y})) \in \mathcal{P}$ is scaled by $s$, which is adaptively determined by class accuracy, and is pasted onto the mapped image to create the final augmented image $(x', \mathbf{y}')$. We aim to maximize the learning of the instance segmentation task through optimizing each stage.

**Quota-Balanced Unique Mapping**    Previous studies [37, 41] adopt a purely random assignment of synthetic objects to real images, $\mathcal{P} = \{(o, (x, \mathbf{y})) \mid \forall (x, \mathbf{y}) \in \mathcal{D}^{real}, o \sim \text{Uniform}(\mathcal{O}^{syn})\}$. While random mapping serves as a strong baseline, it becomes suboptimal under a tight object generation budget. Furthermore, random mapping hinders our class-wise budget optimization since some generated objects may never be selected.

To address these limitations, we introduce *quota-balanced unique mapping*. In this scheme, once an object $o$ is drawn from the synthetic object pool $\mathcal{O}^{syn}$, it is removed so that each synthetic object contributes exactly once toward its class quota. When $\mathcal{O}^{syn}$ is exhausted, it is refilled with the initially generated object pool. This scheme ensures that, across the entire augmentation process, each class is represented according to its budget ratio prescribed in Section 3.3.

**Accuracy-Based Object Resizing**    When pasting an object, existing approaches [37, 41] draw object size from the distribution of each class in the real dataset, i.e., $s \sim \mathcal{N}(\mu_c, \sigma_c^2)$, where $\mu_c$ and $\sigma_c^2$ are the mean and variance of objects of a class $c$ in $D^{real}$. While such an approach helps match the overall scale between synthetic and real objects, it fails to boost learning for rare or low-accuracy classes, since their typically smaller object sizes remain hard to distinguish and learn from.

To remedy this imbalance, we introduce a *resizing* scheme based on the validation accuracy of each class. Specifically, objects from low-accuracy classes are scaled larger to enhance the visibility of their features, thereby facilitating better feature extraction. In contrast, objects from high-accuracy classes are generated at smaller scales to reduce their influence during training and prevent the model from overfitting to already well-represented classes. This simple yet effective strategy balances the contribution of each class by providing richer visual cues for underrepresented classes, enhancing the model's ability to recognize rare objects. The detailed algorithm is provided in Appendix A.3.

## 4 Experiments

### 4.1 Experiment Settings

**Datasets and Implementation Details**    Instance segmentation and object detection experiments are mainly conducted on the LVIS v1.0 dataset [18] as in the relevant literature, with supporting experiments on the Pascal VOC dataset [10] and the Open Images V5 dataset [24]. CenterNet2 [39], with a ResNet-50 [20] backbone, is used as our main instance segmentation model, implemented in Detectron2 [33]. The Swin-L [27] backbone is also used for comparison with baseline methods. Stable Diffusion 3 Medium [9] is used as our generation model, and BiRefNet [38] is used for segmentation on the generated objects. After generation or segmentation, CLIP [29] is used to filter out the low-quality objects whose score is smaller than $0.25$.

Our model is trained for 10 rounds, with each round consisting of 9,000 iterations and a batch size of 16. Our multi-round collaborative augmentation serves as the sole modification to the baseline training pipeline. We evaluate performance using average precision (AP) for both bounding box detection and instance segmentation while also analyzing results across different class frequencies (*rare*, *common*, and *frequent* classes) as defined in LVIS. We use 8 NVIDIA GeForce RTX 3090 GPUs, where 4 GPUs are used for training the model, 3 GPUs for generating objects, and 1 GPU for segmentation. Only for the Swin-L experiment, we use NVIDIA A40 GPUs instead to fit the model in the GPU memory. To run both training and generation without waiting for another process, the number of object generations per round is empirically set to $6 \times 1203$ (number of classes in LVIS). Besides, $\gamma$ and $\omega$ in Eq. (7) are set to 5.0 and 0.03, respectively. Further details about the experiments are provided in Appendix C.

**Compared Methods**    We compare our MRCA with strong generative copy-paste methods, including X-Paste [37], BSGAL [41], and DiverGen [11]. The numerical results of the baselines are borrowed from the BSGAL and DiverGen papers owing to the same experiment settings.

## 4.2 Main Experiment Results

Table 1 and Table 2 show the mean average precision (mAP) on the instance segmentation and object detection tasks on the LVIS v1.0 dataset for the ResNet-50 [20] and Swin-L [27] backbones. The superscripts "box" and "mask" denote the mAP for object detection and instance segmentation, respectively. The subscripts "r", "c", and "f" signify rare, common, and frequent classes, respectively. Overall, MRCA outperforms the state-of-the-art generative augmentation methods while *generating only 6%* as many synthetic objects. These results indeed demonstrate the *sample efficiency* of MRCA, because its multi-round collaborative augmentation delivers the most essential training data at the appropriate time. Furthermore, the performance improvement for rare classes is more pronounced than for other classes, because the rare classes receive superior treatment due to increased generation budgets (§3.3 ) and expanded space (§3.4). See Appendix D for augmented image examples.

Table 1: Comparison with the state-of-the-art methods using the **ResNet-50** backbone.

| Method | # Gen Objects | $AP^{box}$ | $AP^{mask}$ | $AP_r^{box}$ | $AP_r^{mask}$ | $AP_c^{box}$ | $AP_c^{mask}$ | $AP_f^{box}$ | $AP_f^{mask}$ |
|---|---|---|---|---|---|---|---|---|---|
| No Aug. | 0 | 31.50 | 28.20 | 22.60 | 20.20 | 29.30 | 26.70 | 37.80 | 33.40 |
| X-Paste | 1200k | 34.20 | 30.39 | 24.33 | 22.21 | 33.23 | 29.57 | 39.63 | 34.89 |
| X-Paste + CLIP | 1200k | 34.35 | 30.70 | 25.99 | 24.38 | 32.83 | 29.41 | 39.71 | 34.92 |
| BSGAL | 1200k | 35.40 | 31.56 | 27.95 | 25.43 | 34.14 | 30.56 | 40.07 | 35.37 |
| **MRCA** | **72k** | **35.56** | **31.81** | **28.14** | **25.93** | **34.33** | **30.86** | **40.18** | **35.44** |

Table 2: Comparison with the state-of-the-art methods using the **Swin-L** backbone.

| Method | # Gen Objects | $AP^{box}$ | $AP^{mask}$ | $AP_r^{box}$ | $AP_r^{mask}$ | $AP_c^{box}$ | $AP_c^{mask}$ | $AP_f^{box}$ | $AP_f^{mask}$ |
|---|---|---|---|---|---|---|---|---|---|
| No Aug. | 0 | 47.43 | 42.30 | 41.00 | 36.75 | 47.53 | 43.10 | 50.14 | 43.83 |
| X-Paste | 1200k | 49.57 | 43.85 | 44.87 | 39.66 | 49.74 | 44.64 | 51.46 | 44.82 |
| X-Paste + CLIP | 1200k | 49.80 | 44.51 | 45.28 | 40.62 | 49.33 | 44.96 | **52.30** | **45.72** |
| BSGAL | 1200k | 50.47 | 44.85 | 47.55 | 42.37 | 50.43 | 45.47 | 51.79 | 45.26 |
| DiverGen | 1200k | 51.24 | 45.48 | 50.07 | 45.85 | 51.33 | 45.83 | 51.64 | 44.96 |
| **MRCA** | **72k** | **51.80** | **45.91** | **51.58** | **46.84** | **51.86** | **46.31** | 51.84 | 45.05 |

## 4.3 Ablation Study and Further Analysis

**Effect of the Feedback on Generation and Map-Paste**   The ablation study is conducted on the feedback components: class-wise budget optimization, model gradient-based generation, and object resizing. We apply each component individually and collectively in a multi-round manner to confirm its efficacy. Table 3 shows that each individual component contributes to performance gain without interfering with other components.

Table 3: Ablation on the components of MRCA.

| Class Budget | Model Gradient | Resizing | $AP^{box}$ | $AP^{mask}$ | $AP_r^{box}$ | $AP_r^{mask}$ |
|---|---|---|---|---|---|---|
| ✓ | | | 34.86 | 31.33 | 27.47 | 24.91 |
| | ✓ | | 35.04 | 31.14 | 27.13 | 24.28 |
| | | ✓ | 34.91 | 31.10 | 27.52 | 24.89 |
| ✓ | ✓ | | 35.15 | 31.42 | 26.54 | 24.35 |
| ✓ | ✓ | ✓ | **35.56** | **31.81** | **28.14** | **25.93** |

**Effect of the Class-Wise Budget Optimization Components**   We also conduct a fine-grained ablation study on class-wise budget optimization while keeping other main methods (model gradient and resizing) enabled, as shown in Table 4. When using only the number of objects, we observe notable improvements for rare classes.

Table 4: Ablation on the components of class-wise budget optimization.

| Number of Objects ($S$) | Validation Accuracy ($A$) | Classifier Weight ($C$) | $AP^{box}$ | $AP^{mask}$ | $AP_r^{box}$ | $AP_r^{mask}$ |
|---|---|---|---|---|---|---|
| | | | 35.11 | 31.35 | 27.36 | 25.02 |
| ✓ | | | 35.33 | 31.60 | 28.05 | 25.87 |
| ✓ | ✓ | | 35.51 | 31.79 | 27.93 | 25.67 |
| ✓ | ✓ | ✓ | **35.56** | **31.81** | **28.14** | **25.93** |

The validation accuracy term further supports common and frequent classes with low accuracy. Nonetheless, some low-accuracy classes continue to produce similar objects across rounds. To mitigate this issue, we incorporate the classifier weight, which reflects whether gradient feedback is being updated over rounds. As evidenced by the improvements in the result, both validation accuracy and classifier weight contribute effectively as intended.

**Effect of the Generation Model**   In Table 5, we compare the performance when using different generative models, Stable Diffusion 1.5 [30] and Stable Diffusion 3 Medium [9]. The hyperparameters of the diffusion models are set equal, with classifier guidance scale as 5.0, generation step as 30, and image resolution as $512 \times 512$.

Table 5: Effect of the generation model.

| Method | $AP^{box}$ | $AP^{mask}$ | $AP_r^{box}$ | $AP_r^{mask}$ |
|---|---|---|---|---|
| BSGAL (SD 1.5) | 34.82 | 31.21 | 26.76 | 24.84 |
| MRCA (SD 1.5) | 35.12 | 31.38 | 27.42 | 25.21 |
| MRCA (SD 3) | **35.56** | **31.81** | **28.14** | **25.93** |

The recent version of Stable Diffusion generates higher-quality objects, leading to better results. When using the same version, Stable Diffusion 1.5, the superiority of MRCA over BSGAL is still maintained in terms of the mAP for all and rare classes.

**Effect of the Number of Rounds**    We investigate the effectiveness of applying feedback in a *multi-round* manner in Table 6. The number of training rounds is adjusted from 1 to 30, given a fixed number of total training iterations. As shown in Theorem 3.3, when the number of rounds increases from 1 to 10, the quality of instance segmentation and object detection improves, indicating that the quality of the augmentation improves accordingly. Over 10 rounds, however, the performance drops because the model does not learn sufficiently between the short training rounds; that is, a single round is unable to cover a whole epoch over in such conditions.

Table 6: Effect of the number of rounds with a fixed number of total training iterations.

| # of Rounds | # Gen Objects | $AP^{box}$ | $AP^{mask}$ | $AP^{box}_r$ | $AP^{mask}_r$ | $AP^{box}_c$ | $AP^{mask}_c$ | $AP^{box}_f$ | $AP^{mask}_f$ |
|---|---|---|---|---|---|---|---|---|---|
| 1 | 72k | 34.79 | 31.17 | 27.64 | 24.63 | 33.11 | 30.09 | 39.81 | 35.25 |
| 2 | 72k | 34.96 | 31.09 | 27.33 | 24.35 | 33.28 | 29.90 | 40.19 | 35.39 |
| 5 | 72k | 35.09 | 31.38 | 26.16 | 24.04 | 33.76 | 30.44 | **40.39** | **35.54** |
| 10 | 72k | **35.56** | **31.81** | **28.14** | **25.93** | **34.33** | **30.86** | 40.18 | 35.44 |
| 20 | 72k | 34.96 | 31.46 | 27.81 | 24.85 | 33.48 | 30.55 | 39.76 | 35.38 |
| 30 | 72k | 34.81 | 31.13 | 26.86 | 24.49 | 33.45 | 30.14 | 39.82 | 35.15 |

**Effect of the Curriculum Learning**    The proposed MRCA generates objects and trains with the curriculum divided by rounds. To check the effect of the curriculum learning strategy, we separately train a model from scratch given the

Table 7: Effect of the curriculum learning.

| Method | $AP^{box}$ | $AP^{mask}$ | $AP^{box}_r$ | $AP^{mask}_r$ |
|---|---|---|---|---|
| MRCA (w.o. curriculum) | 35.23 | 31.27 | **28.28** | 25.46 |
| MRCA (w. curriculum) | **35.56** | **31.81** | 28.14 | **25.93** |

objects generated over all rounds of MRCA. Table 7 shows the difference depending on the existence of the curriculum learning. Although training without the curriculum has more object quantity at the beginning, the performance becomes lower in most cases. This result suggests that while the objects generated by MRCA can be used for the input to any training pipeline, the optimal strategy is to use the curriculum learning strategy provided by our method.

**Analysis on the Generation Budget**    In previous experiments, we empirically chose a generation budget $B$ of 72k to efficiently pipeline our generation and training processes, ensuring no bubbles. This additional test examines the

Table 8: Analysis on the generation budget.

| # Gen Objects | Training Time | $AP^{box}$ | $AP^{mask}$ | $AP^{box}_r$ | $AP^{mask}_r$ |
|---|---|---|---|---|---|
| 36k | 30h | 34.82 | 31.21 | 26.76 | 24.84 |
| 72k | 34h | 35.56 | 31.81 | 28.14 | 25.93 |
| 144k | 63h | **35.58** | **31.84** | **28.51** | **25.99** |

effect of larger budgets on the mAP as well as the training time excluding the warm-up stage. Table 8 shows that the training time is proportional to the number of object generations, because multi-round training needs to wait for the object generation to be used for augmentation. Only a small performance improvement is observed with an increased generation budget, decreasing the sample efficiency. This limitation of low scalability is further discussed in Appendix E.

**Analysis on the Feedback Guidance Scale**    The feedback guidance scale, $\omega$ in Eq. (7), has a significant impact on the image quality. Table 9 supports that feedback guidance can largely affect the generated image quality. The default value $\omega = 0.03$ usually offers the best performance. The performance drops significantly for

Table 9: Analysis on the feedback guidance scale.

| $\omega$ | $AP^{box}$ | $AP^{mask}$ | $AP^{box}_r$ | $AP^{mask}_r$ |
|---|---|---|---|---|
| 0.01 | 34.87 | 30.95 | 26.90 | 23.83 |
| **0.03** | **35.56** | **31.81** | **28.14** | **25.93** |
| 0.05 | 34.78 | 31.15 | 27.34 | 25.21 |
| 0.10 | 34.25 | 30.42 | 23.98 | 22.93 |

a large $\omega$ because excessively large model guidance overwhelms the trained diffusion denoising gradients. Its effective range can be quickly identified with qualitative analysis, as a high $\omega$ produces unnatural artifacts, while a low $\omega$ results in images nearly identical to unguided outputs.

**Analysis on the Classifier Guidance Scale**    Table 10 presents the effect of the classifier guidance scale, $\gamma$ in Eq. (7). A higher $\gamma$ amplifies the influence of the conditioning signal (e.g., text prompt or class label) on the generation process, which strengthens fidelity and prompt alignment but pushes samples toward a narrower region of

Table 10: Analysis on the classifier guidance scale.

| $\gamma$ | $AP^{box}$ | $AP^{mask}$ | $AP^{box}_r$ | $AP^{mask}_r$ |
|---|---|---|---|---|
| 3.0 | 35.21 | 31.35 | 26.59 | 24.33 |
| **5.0** | **35.56** | **31.81** | **28.14** | **25.93** |
| 7.0 | 35.32 | 31.58 | 26.89 | 24.68 |

the data distribution, thus reducing diversity among outputs. Conversely, when $\gamma$ is too low, the diffusion model behaves closer to its unconditional form. Therefore, the generated samples exhibit higher variability but lower fidelity and semantic accuracy, leading to visibly degraded quality. The default value of 5.0, which has been adopted in prior studies [37, 41], is also found to be effective in our experiments by stabilizing the tradeoffs.

**Experiments on Additional Datasets** We conduct only object detection on Pascal VOC 2012 [10] because the annotations only support semantic segmentation. For Open Images V5 [24], due to the large size of the dataset, we use a pareto sampling method from previous work [32] to create its long-tailed version.

Table 11: Object detection on VOC 2012 and instance segmentation on Open Images V5 long-tail.

| Method | Pascal VOC 2012 | | | Open Images V5 (long tail) | |
| --- | --- | --- | --- | --- | --- |
| | $AP^{50}$ | $AP^{75}$ | AP | $AP^{box}$ | $AP^{mask}$ |
| No Aug. | 68.45 | 49.35 | 44.89 | 45.97 | 36.53 |
| **MRCA** | **72.16** | **55.08** | **50.85** | **47.10** | **37.30** |

Again, MRCA boosts the performance through our novel data augmentation pipeline in these datasets.

### 4.4 Qualitative Analysis

**Round-Wise Object Visualization and Entropy** Figure 4 visualizes a portion of generated objects and their average entropy for each round. As shown in Figure 4a, the generated objects maintain their class characteristics over all rounds. We assess the entropy of objects with respect to two instances of the model: (1) the warmed-up model instance, which is trained solely on real training data, and (2) the feedback model instance, which is the model that guides the generation of the synthetic images in the corresponding round. Given that the guidance scale is constant, the entropy (uncertainty) respect to the feedback model $M_{\phi_r}$ remains similar across rounds, as shown in Figure 4b. However, from the perspective of the initial model $M_{\phi_0}$, the feedback received from more advanced models in subsequent rounds results in the generation of increasingly uncertain or challenging objects, as depicted in Figure 4c. This tendency is also evident when examining the average entropy across all classes in Figure 4d. Overall, MRCA forms an effective easy-to-hard curriculum.

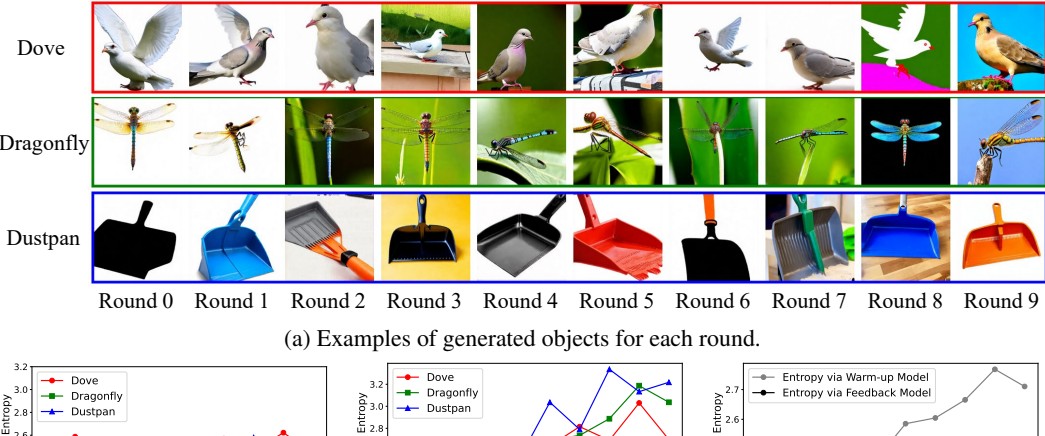

(a) Examples of generated objects for each round.

(b) Mean entropy of objects respect to the feedback model ($M_{\phi_r}$).

(c) Mean entropy of objects respect to the warm-up model ($M_{\phi_0}$).

(d) Class average entropy respect to the warm-up and feedback models.

Figure 4: Examples of generated objects for each round **(a)**, with their class-wise entropy respect to the feedback model **(b)**, the warm-up model **(c)**, and their comparison over average of all classes **(d)**.

## 5 Conclusion

The proposed multi-round collaborative augmentation (MRCA) framework enables generating objects that are continuously more informative to the instance segmentation model *with a considerably small amount of data generations* (e.g., 6%). The instance segmentation model consistently guides the diffusion generation process to create objects that elicit greater uncertainty from the model, hence generating more informative samples, while employing class-wise budget optimization to determine the appropriate number of generations for each class. Then, accuracy-based scaling effectively places the generated objects in real images to focus on learning of rare classes. Experiment results demonstrate that MRCA outperforms the state-of-the-art generative copy-paste augmentation methods for instance segmentation. In conclusion, we assert that the MRCA framework presents a highly promising avenue for enhancing the performance of instance segmentation.

## Acknowledgements

This work was supported by Institute of Information & Communications Technology Planning & Evaluation (IITP) grant funded by the Korea government (MSIT) (No. RS-2020-II200862, DB4DL: High-Usability and Performance In-Memory Distributed DBMS for Deep Learning, 50% and No. RS-2025-25410841, Beyond the Turing Test: Human-Level Game-Playing Agents with Generalization and Adaptation, 50%).

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

# A  Algorithm Pseudocode

## A.1  Multi-Round Collaborative Augmentation Pipeline

The overall procedure of MRCA is explained in Algorithm 1, composed of two phases: the warm-up phase and the multi-round phase. At the warm-up phase, the instance segmentation model $M_\phi$ is trained with real training data in parallel with the generation of objects by $G$. When both processes terminate, the multi-round phase starts. Similar to the warm-up phase, the multi-round phase also undergoes parallel training and generation. The difference lies in applying the feedback from the previous round's model to both object generation and training data augmentation. For the final round, the object generation does not proceed since there are no more training rounds left to use them. Eventually, MRCA returns the trained instance segmentation model and the set of generated objects.

---

**Algorithm 1** Multi-round collaborative augmentation pipeline

---

1: **Given** instance segmentation model $M_{\phi_r}$ parametrized by $\phi$ at a round $r$ that creates feedback $f_{\phi_r}$, object generator $\mathcal{G}$, real training data $\mathcal{D}_0 = \mathcal{D}^{real}$, total round $N$
2: **Do in parallel**                                                            ▷ Warm-up phase
3:     Train $M_{\phi_{init}}$ on $\mathcal{D}_0 \to M_{\phi_0}$
4:     Generate objects by $\mathcal{G}$ with no feedback $\to \mathcal{O}_1^{syn}$
5: **end**
6: **for** $r = 1$ **to** $N$ **do**                                             ▷ Multi-round phase
7:     **Do in parallel**
8:         Augment $\mathcal{D}_0$ with $\cup_{i=1}^r \mathcal{O}_i^{syn}$ and feedback $f_{\phi_{r-1}} \to \mathcal{D}_r$, Train $M_{\phi_{r-1}}$ on $\mathcal{D}_r \to M_{\phi_r}$
9:         **if** $r \neq N$ **then**
10:             Generate objects by $\mathcal{G}$ with feedback $f_{\phi_{r-1}}$ from $M_{\phi_{r-1}} \to \mathcal{O}_r^{syn}$
11:         **end if**
12:     **end**
13: **end for**
14: **Return** trained instance segmentation model $M_{\phi_N}$, generated objects $\cup_{i=1}^N \mathcal{O}_i^{syn}$

---

## A.2  Diffusion with Feedback from the Instance Segmentation Model

The feedback-guided diffusion-based object generation is explained by two hierarchical algorithms. At a higher level, Algorithm 2 describes where the feedback is applied in the diffusion process. A general diffusion process is composed of multiple denoising steps, where the scheduled denoising is applied sequentially. We apply the feedback guidance selectively after the scheduled denoising to keep the uncertainty in range so that the generated images remain in the context of the input class. In our experiments, the feedback guidance is applied every 5 steps out of 30 steps, for a total of 6 times per image. The hyperparameter $\omega$ in Line 6 is set to 0.03.

---

**Algorithm 2** Denoising the diffusion model with a criterion function

---

1: **Given** stable diffusion model latent state $z_t$, denoising U-Net $\epsilon_\theta$ with scheduler $\sigma(t)$, conditioning $\tau_\theta$, stable diffusion vae decoder $v_{dec}$, instance segmentation model $M_\phi$, scale of feedback guidance $\omega$, and criterion function $\mathcal{C}$
2: **for** each step $t$ from $T$ **to** 1 **do**
3:     $\hat{\epsilon} = \epsilon_\theta(z_t, t, \tau_\theta)$                                 ▷ Predict noise
4:     $z'_{t-1} = z_t - \sigma(t)\hat{\epsilon}$                             ▷ Denoise according to the scheduler
5:     **if** $t$ is step for object detector feedback **then**
6:         $z_{t-1} = z'_{t-1} - \omega\mathcal{C}(z'_{t-1}, M_\phi, v_{dec})$     ▷ Apply instance segmentation model's feedback
7:     **else**
8:         $z_{t-1} = z'_{t-1}$
9:     **end if**
10: **end for**
11: **Return** denoised latent state $z_0$

---

Algorithm 3 explains in more detail how the instance segmentation model and the criterion function compute the gradients for the diffusion model. The key idea is setting the whole image as the object proposal for extracting the features, since we generate a single object per image. Lines 2–5 explain the criterion function computation line by line.

---

**Algorithm 3** Applying the criterion function to the instance segmentation model

---

1: **Given** intermediate stable diffusion model latent state $z'_{t-1}$, stable diffusion vae decoder $v_{dec}$, instance segmentation model $M_\phi$ consisting of backbone(feature extractor) $b_\phi$ and classifier $c_\phi$, and criterion function $\mathcal{C}$
2: $\tilde{x}_{t-1} = v_{dec}(z'_{t-1})$ $\qquad\qquad\qquad\qquad\qquad$ ▷ Create an image from latent variable
3: $feat = b_\phi(\tilde{x}_{t-1})$ $\qquad\qquad\qquad\qquad$ ▷ Extract features from detector's backbone
4: $logits = c_\phi(feat(prop))$ $\qquad$ ▷ With whole image as proposal, classify corresponding features
5: $grads = \mathcal{C}(logits, z'_{t-1})$ $\qquad\qquad\qquad$ ▷ Calculate gradients respect to criterion function
6: **Return** gradients on latent state $grads$

---

## A.3 Object Resizing Based on Class-Wise Evaluation

Algorithm 4 provides a more detailed explanation of the resizing method. Given the average precision $A_c$ of a specific class, we assign a scale factor to a class, which will cause its objects to be resized in accordance with the scale factor. For three values of $A_c$, 0%, $\bar{A}$ (mean), and 100%, $A_c = 0\% \rightarrow scale = s_{max}$, $\quad A_c = \bar{A} \rightarrow scale = 1$, $\quad A_c = 100\% \rightarrow scale = \frac{1}{s_{max}}$. For other values of $A_c$, we linearly interpolate in log scale.

---

**Algorithm 4** Computing a scale factor based on average precision (AP)

---

1: **function** COMPUTE_SCALE_FACTOR($A_c, \bar{A}$)
2: $\quad$ **Input:** $A_c$ (Average precision of a class $c$), $\bar{A}$ (Mean average precision of all classes), $s_{max}$ (Maximum scale)
3: $\quad$ **Output:** Computed scale factor
$\qquad\qquad\qquad\qquad\qquad\qquad\qquad\qquad\qquad\qquad$ ▷ Log scale values for interpolation
4: $\quad$ log_scale_min $\leftarrow \ln(s^2_{max})$ $\qquad\qquad\qquad\qquad\qquad$ ▷ Scale at $A_c = 0\%$
5: $\quad$ log_scale_max $\leftarrow \ln\left(\frac{1}{s^2_{max}}\right)$ $\qquad\qquad\qquad\qquad$ ▷ Scale at $A_c = 100\%$
$\qquad\qquad\qquad\qquad\qquad\qquad\qquad\qquad$ ▷ Piecewise linear interpolation in log scale
6: $\quad$ **if** $A_c < \bar{A}$ **then**
7: $\qquad$ log_scale $\leftarrow$ log_scale_min $\left(\frac{\bar{A}-A_c}{\bar{A}}\right)$
8: $\quad$ **else**
9: $\qquad$ log_scale $\leftarrow$ log_scale_max $\left(\frac{A_c-\bar{A}}{100-\bar{A}}\right)$
10: $\quad$ **end if**
11: $\quad$ **return** scale = $\exp($log_scale$)$
12: **end function**

---

# B Proof of Theorem 3.3

We begin by expressing the quality of the final dataset in the multi-round setting

$$Q(\mathcal{D}_{N,B}) = Q\left(\mathcal{A}_{\frac{1}{N}B}(\mathcal{D}_0; f_{\phi_{r-1}}, \cup_{i=1}^{r-1}\mathcal{O}_i^{\text{syn}})\right). \tag{8}$$

Consider the $\frac{1}{N}B$ objects created at a round $r$. Since the feedback from the recent model $f_{\phi_{r-1}}$ is more informative than $f_{\phi_{r-2}}$, $\mathcal{I}_\mathcal{D}(M_{\phi_{r-1}}) \geq \mathcal{I}_\mathcal{D}(M_{\phi_{r-2}})$, by Lemma 3.2,

$$Q\left(\mathcal{A}_{\frac{1}{N}B}(\mathcal{D}_0; f_{\phi_{r-1}}, \cup_{i=1}^{r-1}\mathcal{O}_i^{\text{syn}})\right) \geq Q\left(\mathcal{A}_{\frac{2}{N}B}(\mathcal{D}_0; f_{\phi_{r-2}}, \cup_{i=1}^{r-2}\mathcal{O}_i^{\text{syn}})\right). \tag{9}$$

Similarly, continuing for $\frac{1}{N}B$ objects at rounds $r-1, r-2, \cdots, 1$,

$$Q\left(\mathcal{A}_{\frac{2}{N}B}(\mathcal{D}_0; f_{\phi_{r-2}}, \cup_{i=1}^{r-2}\mathcal{O}_i^{\text{syn}})\right) \geq Q\left(\mathcal{A}_{\frac{3}{N}B}(\mathcal{D}_0; f_{\phi_{r-3}}, \cup_{i=1}^{r-3}\mathcal{O}_i^{\text{syn}})\right) \tag{10}$$

$$\cdots$$
$$Q\left(\mathcal{A}_{\frac{N-1}{N}B}(\mathcal{D}_0; f_{\phi_1}, \mathcal{O}_1^{\mathrm{syn}})\right) \geq Q\left(\mathcal{A}_{\frac{N}{N}B}(\mathcal{D}_0; f_{\phi_0}, \emptyset)\right). \tag{11}$$

By multiplying all inequalities from Eq. (9) to Eq. (11),

$$Q\left(\mathcal{A}_{\frac{1}{N}B}(\mathcal{D}_0; f_{\phi_{r-1}}, \cup_{i=1}^{r-1}\mathcal{O}_i^{\mathrm{syn}})\right) \geq Q\left(\mathcal{A}_{\frac{N}{N}B}(\mathcal{D}_0; f_{\phi_0}, \emptyset)\right) \tag{12}$$

$$\Rightarrow Q(\mathcal{D}_{N,B}) \geq Q(\mathcal{D}_{1,B}). \tag{13}$$

## C  More Details for Experiments

**Object Edge Smoothing**    When pasting objects, we do not employ any methods for smoothing object edges, such as Gaussian blurring or alpha/Poisson blending. Based on previous work [15] and our own evaluation, the edge handling methods have no significant effect on the overall performance.

**Reason for Using the ResNet Backbone**    Since the overall performance trend remains consistent across different backbone models (ResNet and Swin-L), we choose a lighter model to enable more extensive ablation studies within a given time. Notably, the performance improvement reported in Table 1 and Table 2 is even larger when using the Swin-L backbone, supporting that our ResNet-based ablation results do not overstate the effectiveness of MRCA. The ResNet backbone is more memory-efficient and can be trained on a 24GB RTX 3090, whereas the Swin-L backbone requires a higher-memory setup, such as a 48GB A6000 or larger. We believe that both models are important, as they represent a trade-off between model size and performance, which is an important consideration for practical deployment, particularly in on-device scenarios.

**Overlap of Augmented Objects**    The augmented objects can be overlapped in our experiment setting. When an object is overlapped, its object mask is adjusted to eliminate the obscured portion. This scheme is a common practice in the relevant studies [7, 37, 41].

**Unrealistic Augmented Objects**    Our approach follows the line of research established by X-Paste [37], BSGAL [41], and DiverGen [11], which leverage synthetically generated, often unrealistic, map-pasted images to improve performance. While the augmented images may not share the same distribution as the original images in terms of object size, they still contribute positively to instance segmentation performance by incorporating knowledge from the generation model. The MRCA framework further enhances this effect by explicitly optimizing both object generation and pasting strategies. It is worth noting that the use of visually unrealistic images for data augmentation has been widely observed to be effective in improving downstream performance. For instance, also in image classification, CutMix [35] often produces implausible examples such as images featuring a cat's head on a dog's body.

## D  Visualization of Augmented Images

We visualize some of the augmented images in Figure 5. During training, we paste 1–3 generated objects onto an image from the real LVIS training set. We keep the number of pasted objects small to minimize the deviation in the average number of objects per image compared to real images. The sizes of the objects are managed with our accuracy-based resizing scheme (§3.4). For example, in the first image, the class "cockroach" has low accuracy, and therefore an object of that class is pasted with a larger size compared to its actual size from real images.

## E  Limitations

The current implementation of MRCA shows low scalability, meaning that generating more objects with the current framework does not lead to a large increase in performance. This downside is due to our method's focus on uncertainty, which is effective in generating small-scale datasets since there is a low chance of overlap between objects. However, generating large-scale datasets without a diversity-focused method may generate similar objects, resulting in low scalability. Increasing the diversity of the generated objects effectively addresses the scalability issue, as reducing overlap between objects allows for the acquisition of more information. DiverGen [11], a prior study that focuses on the diversity of the generated dataset, supports our earlier argument with scalable results.

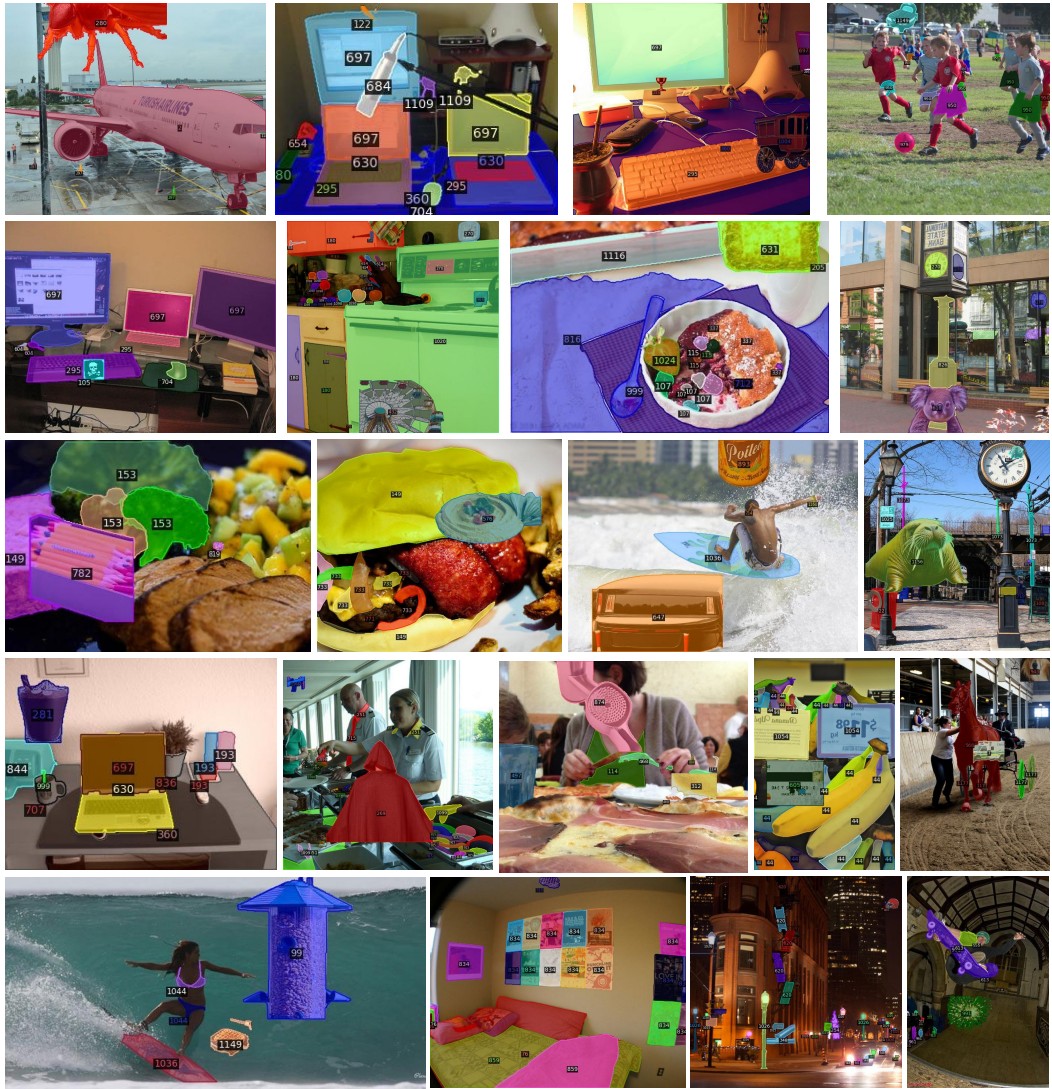

Figure 5: Examples of augmented images.

Furthermore, the current framework only works with a single generator to efficiently use feedback. Previous studies, such as BSGAL [41] and DiverGen [11], demonstrate that using multiple generation models for generating objects can enhance data quality by increasing diversity. Therefore, employing multiple generation models in our multi-round augmentation framework remains a future task. Such an approach can increase diversity in another dimension, as well as solve the scalability issue.

We also note that scaling of rare classes can lead to undesirable effects in domains where object size carries critical contextual meaning. For instance, in medical imaging or other domains with consistent scales, resizing objects may distort important spatial cues and disrupt the natural size distribution. To mitigate such risks, resizing strategies should ideally be avoided or applied only within narrow bounds when using MRCA in these settings. However, in real-world image datasets such as LVIS [18], PASCAL [10], and OpenImages [24], where object scales vary greatly even within the same category, we did not observe notable negative impacts on detection performance. By carefully constraining the maximum scaling coefficient, we ensured that objects did not become unnaturally large or small while maintaining overall realism. Moreover, as the representation of rare classes improves through successive training rounds, their resulting scale distribution tends to align more closely with natural proportions, further reducing the potential adverse effects of scaling.

