# OpenReview forum: "Sample-Efficient Multi-Round Generative Data Augmentation for Long-Tail Instance Segmentation"
_NeurIPS.cc/2025/Conference — NeurIPS 2025 poster_

### Official Review · Reviewer_3V2k · 2025-06-29

**Clarity:** 3
**Significance:** 3
**Originality:** 2
**Rating:** 4
**Confidence:** 4

**Summary:**

This paper proposes a multi-round collaborative augmentation (MRCA) framework for long-tail instance segmentation. By integrating feedback from an instance segmentation model into both object generation and placement, MRCA adaptively synthesizes challenging and informative training samples. It uses diffusion models guided by model uncertainty and class-wise accuracy, and dynamically adjusts generation budgets. Experiments on LVIS show that MRCA achieves superior performance with only 6% of the synthetic data used by prior methods.

**Questions:**

How do you ensure that the augmented data and the original data share the same distribution? From my perspective, the map-pasted pictures seem not quite realistic.

**Ethical Concerns:**

["NO or VERY MINOR ethics concerns only"]

**Final Justification:**

Most of my concerns have been addressed. I especially appreciate it that the authors explain the model sensitivity to hyperparameters and the use of "unrealistic" images. While I still believe that the theoretical analysis is rather superficial and only marginally insightful, the proposed method is in itself a contribution to the field. Overall, I decide to keep my rating at 4.

**Limitations:**

Yes, the authors have adequately addressed the limitations and potential negative societal impact of their work.

**Quality:**

2

**Strengths And Weaknesses:**

## Strengths

1. The paper is well-organised with clear structure.
2. The paper integrates two-fold model feedback (class-wise budget optimization + gradient-guided generation) in a multi-round loop, which is a principled extension of curriculum learning to generative augmentation.
3. The paper effectively disentangles contributions of each feedback component (budget optimization, gradient-based generation, resizing).

## Weaknesses

1. Theoretical analysis is superficial. Assumption 3.1 merely states that feedback from a trained model improves augmentation quality, but does not formalize under what conditions this holds (e.g. model capacity limits, data distribution shifts). Furthermore, The paper references information gain $\mathcal{I}\_{\mathcal{D}}(M\_{\phi})$ but does not define how it is computed or measured in practice.
2. Feedback guidance hyperparameter sensitivity. Performance is sensitive to the feedback guidance scale (ω). While the paper provides analysis, tuning such parameters can be non-trivial in practice.

---

> ### Author Rebuttal · Authors · 2025-07-30
>
> We sincerely appreciate the positive feedback on our work. We hope that our rebuttal has addressed all major concerns and clarified any remaining questions.
>
> `W1. Require more detailed theoretical analysis on information gain and the conditions where Assumption 3.1 holds.`
>
> We fully understand your perspective. Yes, Assumption 3.1 states that feedback from a trained model improves augmentation quality. However, a similar assumption has been **widely adopted in recent data augmentation literature**, where model feedback consistently improves augmentation quality over unguided approaches. This assumption is empirically validated across numerous established methods[1, 2, 3, 5]. Also, the two conditions raised by the reviewer, (1) the model having sufficient capacity to learn from synthetic data and (2) controlled distribution shift between real and augmented data, are naturally met as training converges.
>
> Regarding the information metric $\mathcal{I_D}(M_{\phi})$, it can be any metric that quantifies how well the model $M_{\phi}$ captures the dataset $\mathcal{D}$. In practice, it can be measured via standard metrics like accuracy or as the reciprocal of the KL divergence between the true and model's distributions: $\mathcal{I_D}(M_{\phi})= {1\over{D_{KL}(p_{\mathcal{D}} || p_{M_{\phi}})}}$. A higher $\mathcal{I_D}(M_{\phi})$ means closer alignment to the true data distribution, enabling more effective feedback $f_\phi$ to improve augmentation quality, such as addressing class imbalance or focusing on uncertain examples.
>
> Per your comment, **we will revise our Assumption 3.1** by explicitly stating the required conditions.
>
> > **Assumption 3.1 (revised)**
> > Let $\mathcal{A}$ denote a data augmentation applied to a dataset $\mathcal{D}$, and let $f_\phi$ denote the feedback from a model $M_\phi$ trained on the dataset $\mathcal{D}$. We define the information metric $\mathcal{I_D}(M_{\phi})$ as a measure of how well the model $M_{\phi}$ captures the distribution of the dataset $\mathcal{D}$. In practice, it can be measured by the average accuracy, i.e., $\mathcal{I_D}(M_{\phi})= \frac{1}{|\mathcal{D}|} \sum_{(x, \mathbf{y}) \in \mathcal{D}} \frac{1}{|\mathbf{y}|} \sum_{y \in \mathbf{y}} \mathbf{1}\left[ \exists \hat{y} \in \hat{\mathbf{y}}(x) \text{ s.t. } \text{IoU}(y, \hat{y}) \geq 0.5 \right]$.
> >
> > **($C1$) Model Capacity**: Model capacity can be represented by the empirical risk $\mathcal{R_{emp}}$, which is defined as the average loss $\ell$ over the dataset $\mathcal{D}$. Formally, $\mathcal{R_{emp}}(M_{\phi}) = \frac{1}{|\mathcal{D}|} \sum_{(x_i, \mathbf{y_{\mathcal{i}}}) \in \mathcal{D}} \ell(M_\phi(x_i),\mathbf{y_{\mathcal{i}}})$. To ensure the model retains available learning capacity, the empirical risk should be decreasing over time, i.e., $\frac{d\mathcal{R_{emp}}(M_\phi^{(t)})}{dt} < 0$.
> >
> > Then, $C1 \rightarrow \mathcal{I_D}(M_\phi) \geq \mathcal{I_D}(\emptyset)$.
> >
> > **($C2$) Controlled Distribution Shift**:  The knowledge gained through augmentation should surpass the adverse effects of distribution shift. Formally, given the model's learned distribution $p_{M_{\phi}}^{(t)}$ over time $t$,  $\frac{d{D_{KL}(p_{\mathcal{D}} || p_{M_{\phi}}^{(t)})}}{dt} < 0$.
> >
> > **($C3$) Feedback Quality**: The feedback $f_\phi$ effectively guides the augmentation process toward quality maximization.
> >
> > Then, $[\mathcal{I_D}(M_\phi) \geq \mathcal{I_D}(\emptyset)] \land (C2) \land (C3) \rightarrow Q(\mathcal{A}(\mathcal{D}; f_\phi)) \geq Q(\mathcal{A}(\mathcal{D}))$.
>
> Condition ($C1$) and ($C2$) are typically satisfied as the accuracy saturates during the training. For condition ($C3$), the design of the feedback mechanism $f_\phi$ and how it guides the augmentation process toward quality maximization are detailed in Sections 3.3 and 3.4.
>
> `W2. Highly sensitive feedback guidance scale can be difficult to tune.`
>
> Although the feedback guidance scale $\omega$ may appear sensitive, its effective search space can be quickly narrowed through simple qualitative analysis. When $\omega > 0.10$, the diffusion model fails to produce natural images, often resulting in artifacts such as white blurs and black holes. These artifacts diminish as $\omega$ decreases and become infrequent around $\omega \approx 0.05$. Conversely, for $\omega < 0.01$, the generated images with feedback are nearly indistinguishable from those without feedback, indicating that meaningful improvements only begin to emerge when $\omega \geq 0.01$. Therefore, the practical and effective range for $\omega$ lies between 0.01 and 0.05, which is **tractable in practice**.
>
> `Q1. The map-pasted images do not seem realistic. Do the augmented images and the original image share the same data distribution?`
>
> Our approach follows the line of research established by X-Paste[4], BSGAL[5], and DiverGen[6], which leverage synthetically generated, **often unrealistic**, map-pasted images to improve performance. While the augmented images may not share the same distribution as the original images in terms of object size, they still contribute positively to instance segmentation performance by incorporating knowledge from the generation model. The MRCA framework further enhances this effect by explicitly optimizing both object generation and pasting strategies. It is worth noting that **the use of visually unrealistic images for data augmentation has been widely observed to be effective in improving downstream performance**. For instance, also in image classification, CutMix[7] often produces implausible examples such as images featuring a cat's head on a dog's body.
>
> ---
> Overall, we will incorporate these responses into the final version of the paper. We hope that most of your concerns have been addressed and would be glad to engage in further discussion if needed.
>
> ---
>
> [1] "AdaAug: Learning Class- and Instance-adaptive Data Augmentation Policies", In ICLR, 2022
>
> [2] "Feedback-guided Data Synthesis for Imbalanced Classification", In TMLR, 2024
>
> [3] "Controlled Training Data Generation with Diffusion Models", In TMLR, 2025
>
> [4] "X-paste: Revisiting scalable copy-paste for instance segmentation using clip and stablediffusion", In ICML, 2023
>
> [5] "Generative active learning for long-tailed instance segmentation", In ICML, 2024
>
> [6] "Diversify your vision datasets with automatic diffusion-based augmentation", In CVPR, 2024
>
> [7] "CutMix: Regularization Strategy to Train Strong Classifiers with Localizable Features", In ICCV, 2019

---

> > ### Comment · Reviewer_3V2k · 2025-08-01
> >
> > Thank you for your detailed response! Your reply to `W2` has addressed my concern to hyperparameter sensitivity. While I still believe that the theoretical analysis is rather superficial and only marginally insightful, I acknowledge that the method performs well empirically. After all, what truly matters is whether a method works well in practice. Given that, I’ve decided to keep my score.

---

> > > ### Author Response · Authors · 2025-08-01
> > >
> > > Thank you for your thoughtful follow-up and for taking the time to consider our response. We appreciate your acknowledgement of the empirical strengths of our method and your decision to keep a positive evaluation.
> > >
> > > While we recognize your insights regarding the theoretical analysis, we would like to emphasize that Assumption 3.1, which posits that feedback from a trained model can guide better augmentation, is a widely adopted principle in recent data augmentation literature. Our key contribution lies in demonstrating that leveraging this feedback in a multi-round manner yields even greater benefits. We believe our empirical results support this insight and demonstrate the practical robustness of the method. We remain committed to further strengthening the theoretical foundations in future work and appreciate your constructive feedback as part of that process.

---

### Official Review · Reviewer_SKJ8 · 2025-07-01

**Clarity:** 3
**Significance:** 3
**Originality:** 3
**Rating:** 4
**Confidence:** 2

**Summary:**

This paper introduces Multi-Round Collaborative Augmentation (MRCA), a novel generative data augmentation framework designed to enhance sample efficiency for long-tail instance segmentation tasks. MRCA addresses the inefficient utilization of pre-generated data by incorporating feedback from an instance segmentation model to guide the augmentation process. It dynamically adjusts the number and size of synthetic objects for each class based on the model's state, improving learning for underrepresented classes. This multi-round process allows feedback to be refined throughout training , enabling MRCA to outperform state-of-the-art methods while requiring only 6% of their data generation.

**Questions:**

**Effectiveness of Guidance from Noisy Intermediate States:** The paper's method applies guidance from the instance segmentation model (Mϕ) to noisy intermediate diffusion states. A key question is whether this guidance remains effective early in the diffusion process, when latent states are still heavily dominated by noise. Features extracted from such noisy representations might be unreliable, potentially leading to suboptimal or counterproductive signals for guiding object generation.

**Ethical Concerns:**

["NO or VERY MINOR ethics concerns only"]

**Limitations:**

yes

**Quality:**

3

**Strengths And Weaknesses:**

strengths

1. **Methodological Soundness and Intuitive Design:** The proposed Multi-Round Collaborative Augmentation (MRCA) framework is conceptually strong and highly intuitive. By incorporating feedback from the instance segmentation model to guide the generative process, MRCA ensures that synthetic data is optimally tailored to the model's evolving needs, which makes inherent sense for improving learning efficiency. The dynamic adjustment of object generation and resizing based on model uncertainty and class accuracy directly targets areas of weakness, providing a logically sound approach to data augmentation.
2. **Empirical Validation and Significant Performance Gains:** The paper provides compelling experimental evidence demonstrating MRCA's effectiveness. Notably, MRCA outperforms state-of-the-art generative augmentation methods on the LVIS v1.0 dataset while requiring only 6% of the data generation. This substantial reduction in generated objects coupled with superior performance, particularly in challenging rare classes, clearly validates the method's sample efficiency and practical utility.
3. **Exemplary Presentation and Clarity:** The paper is exceptionally well-structured and clearly written, enhancing its readability and impact. The figures, such as Figure 1 illustrating the novelty of MRCA and Figure 4 visualizing round-wise object generation and entropy, are highly informative and visually clear, effectively conveying complex ideas. The use of tables to present detailed experimental results and ablation studies further contributes to the paper's comprehensiveness and allows for easy understanding of the contributions of individual components.

weakness

1. **Limited Impact of Core Feedback Mechanisms Compared to Simpler Design:** Table 3  in the ablation study suggests that the more complex feedback mechanisms, such as "Class Budget" and "Model Gradient," contribute less to overall performance improvement than the seemingly simpler "Resizing" component. This observation raises questions about the relative efficacy and necessity of intricate feedback loops, as the straightforward resizing based on accuracy  appears to be a more dominant factor in the gains. Further analysis or alternative designs for the feedback-guided generation could be explored to maximize the impact of these sophisticated components.
2. **Insufficient Granularity in Ablation of Equation (6) Components:** The paper's current ablation study does not fully explore the individual contributions of the terms within Equation (6), which governs the Class-Wise Budget Optimization. Specifically, it would be beneficial to ablate the impact of including or excluding components like the classifier weight difference term and the validation accuracy . A more detailed breakdown would provide clearer insights into which specific feedback signals are most influential for dynamic budget allocation and why, thereby strengthening the empirical foundation of this crucial part of the methodology.

---

> ### Author Rebuttal · Authors · 2025-07-30
>
> We sincerely appreciate the positive feedback on our work. We hope that our rebuttal has addressed all major concerns and clarified any remaining questions.
>
> `W1. Is resizing's effect much larger than complex class budget and model gradient?`
>
> We apologize for any confusion caused by our presentation. The contributions of the three components, "Class Budget," "Model Gradient," and "Resizing," are comparable in magnitude. Since the result using only the "Resizing" component was omitted from Table 3 in the original submission, we have now included it in Table R1. As shown in Table R1, **the "Resizing" component alone does *not* yield a dominant performance gain**, which supports our claim that **a sophisticated feedback-guided generation mechanism is essential**. The best performance is achieved through the synergistic integration of all three components.
>
> Table R1: Ablation on the main components of MRCA using LVIS on ResNet-50. **The 3rd row is added to Table 3 of the original submission.**
> | Class Budget | Model Gradient | Resizing |   $\mathbf{AP^{box}}$  | $\mathbf{AP^{mask}}$  |  $\mathbf{AP^{box}_r}$ | $\mathbf{AP^{mask}_r}$ |
> |:------------:|:------------:|:------------:|:----------------:|:---------------:|:----------:|:----------:|
> | $\checkmark$    | |  |    34.86 | 31.33 |27.47 | 24.91
> |    |$\checkmark$ |  |    35.04 | 31.14 | 27.13 | 24.28
> |      |  | $\checkmark$ |   34.91 |  31.10 |  27.52 |  24.89
> |  $\checkmark$     | $\checkmark$  |  |   35.15 |  31.42 |  26.54 |  24.35
> |  $\checkmark$  |  $\checkmark$ | $\checkmark$ |  **35.56** | **31.81** | **28.14** | **25.93**
>
>
> `W2. Insufficient granularity in ablation of class budget optimization (Equation (6)).`
>
> We conducted a **fine-grained** ablation study on class-wise budget optimization while keeping other components enabled, as shown in Table R2. Equation (6) incorporates three types of information: the number of objects ($S$), validation accuracy ($A$), and classifier weight ($C$). When using only the number of objects, we observed notable improvements for **rare** classes (see the difference in $\mathbf{AP^{box}_r}$ and $\mathbf{AP^{mask}_r}$ between the first and second rows). Then, to better support **common** and **frequent** classes with low accuracy, we introduced the validation accuracy. Nonetheless, some low-accuracy classes continued to produce similar objects across rounds. To mitigate this issue, we incorporated the classifier weight, which reflects whether gradient feedback is being updated over rounds. As evidenced by the improvements in $\mathbf{AP^{box}}$ and $\mathbf{AP^{mask}}$ in the third and fourth rows, both validation accuracy and classifier weight contribute effectively as intended.
>
> Table R2: **Fine-grained** ablation on the components of class budget optimization using LVIS on ResNet-50.
> | Number of objects($S$) | Validation accuracy($A$) | Classifier weight($C$) |   $\mathbf{AP^{box}}$  | $\mathbf{AP^{mask}}$  |  $\mathbf{AP^{box}_r}$ | $\mathbf{AP^{mask}_r}$ |
> |:------------:|:------------:|:------------:|:----------------:|:---------------:|:----------:|:----------:|
> |   | |      | 35.11  | 31.35  | 27.36 |  25.02
> | $\checkmark$  | |      | 35.33   | 31.60  | 28.05 | 25.87
> | $\checkmark$  |$\checkmark$  |   | 35.51    | 31.79  | 27.93 | 25.67
> | $\checkmark$ |$\checkmark$  |   $\checkmark$    |  **35.56** | **31.81** | **28.14** | **25.93**
>
> `Q1. Effectiveness of guidance from noisy intermediate states.`
>
> Thank you for the insightful comment. We indeed explored **three** different strategies for applying feedback guidance from the instance segmentation model to the diffusion model.
>
> 1. In the selected method, feedback guidance is **applied every 5 steps out of a total of 30 diffusion steps** (i.e., 6 times in total). This method was empirically found to offer the best trade-off between performance and image quality.
> 1. The next method is exactly what you mentioned. **Feedback guidance is applied only during the last 6 steps**. While this method reduces the influence of noisy intermediate features, we found that applying guidance in consecutive steps disrupted the denoising process, leading to degraded image quality.
> 1. Another method involves performing an approximate **single-step denoising from an intermediate noisy state to the final state**, followed by extracting features as described in [1]. However, this method significantly increased generation time and yielded negligible performance improvements.
>
> ---
> Overall, we will incorporate these responses and additional results into the final version of the paper. We hope that most of your concerns have been addressed and would be glad to engage in further discussion if needed.
>
> ---
> [1] "Test-time Alignment of Diffusion Models without Reward Over-optimization", In ICLR, 2025.

---

> > ### Author Response · Authors · 2025-08-05
> >
> > We sincerely thank you for the time and effort invested in reviewing our paper. In our rebuttal, we have clarified the necessity of the core feedback mechanism, provided a fine-grained ablation study, and elaborated on the feedback guidance method. We believe these additions address your concerns and further strengthen the contributions of our work. Thank you once again for your thoughtful feedback. Please feel free to reach out with any further questions, and we would be happy to continue the discussion.

---

> ### Author Response · Authors · 2025-08-08
> **Kindly Follow-Up: Reviewer SKJ8**
>
> To the Reviewer, SKJ8
>
> Thank you for reviewing our work and providing insightful comments that have improved its quality significantly. This is a kind reminder that the author/reviewer discussion session is about to finish in only one day (24 hours). We have provided a response to your questions, which includes the class-wise budget optimization and resizing ablation experiments. We believe our response has provided sufficient clarification.  If you need more information or have any other questions, please let us know.
>
>
> Warm regards,
>
> Authors.

---

### Official Review · Reviewer_CkKJ · 2025-07-02

**Clarity:** 3
**Significance:** 3
**Originality:** 3
**Rating:** 4
**Confidence:** 4

**Summary:**

This manuscript presents a collaborative, multi-round approach (MCRA) using feedback from an instance segmentation model, to guide data augmentation after synthesis. This method is claimed to require only a small fraction (about 6%) of the amount of synthetic generated data compared to other current methods, while improving average precision for bounding box and instance segmentation.

**Questions:**

1. In Section 3.1, it might be clarified as to whether the same generated image (from the frozen generator $G$) may be associated with different segmentation $S$ in different rounds, due to the effect of the evolving segmentation model.

2. In Section 3.1, it might be clarified as to whether there is any processing/blurring (gradient?) of object edges, for the Map-Paste() function.

3. In Section 3.2, the proof of Lemma 3.2 (i.e. "increased information" of D necessarily guides augmentation of higher quality) appears somewhat informal. The concept of "information" applied here, and how increased dataset information implies improved augmentation, could be explained or derived in greater detail.

4. In Section 3.2, "as round pass" might be "as rounds pass".

5. In Section 3.3, it is stated that entropy is used as the criterion to steer diffusion sampling. It might be clarified as to how entropy is measured for data instances (also with respect to the warmup and feedback models, as explored in Section 4.4).

6. From Section 4.2 (Tables 1 and 2), the Swin-L backbone significantly outperforms the ResNet-50 backbone. Given this, was there a reason to use the ResNet-50 backbone as the baseline main instance segmentation model, since that could exaggerate the potential gains from MCRA in practical use?

7. For the augmentation examples in Appendix C, it might be clarified as to whether augmented objects can be overlapped.

**Ethical Concerns:**

["NO or VERY MINOR ethics concerns only"]

**Final Justification:**

The authors have addressed all points raised in the original review, with the additional ablation studies appreciated. The concerns on generalizability nonetheless remain. The justification for ResNet versus Swin-L as backbone moreover suggests that the impact of the proposed MCRA method may be diminished with stronger base models. Nonetheless, it remains a useful empirical technique for consideration.

**Limitations:**

yes

**Quality:**

3

**Strengths And Weaknesses:**

Strengths
 - (Quality) Comprehensive ablation experiments isolating effects of feedback, generation model, curriculum learning, guidance scale and generation budget
 - (Originality) Integrates well-known ideas on model feedback and guided learning to the segmentation task


Weaknesses
 - (Quality) Ablation on classifier guidance appears missing
 - (Significance) While the requirement for fewer synthetic examples is claimed as a major benefit of MCRA, no ablation analysis on the effect of reduced synthetic examples (i.e. 72k instead of 1200k) is performed for the competing methods; this is important as Table 8 suggests potential early saturation of the impact of additional synthetic examples
 - (Significance) Performance improvements appear task-specific, and findings (e.g. on optimal hyperparameters) may not be fully generalizable

---

> ### Author Rebuttal · Authors · 2025-07-30
>
> We sincerely appreciate the positive feedback on our work. We hope that our rebuttal has addressed all major concerns and clarified any remaining questions.
>
> `W1. Ablation on classifier guidance.`
>
> **We have conducted an ablation study on the classifier guidance ($\gamma$) in Table R1** by adding the results with $\gamma$=3.0 and $\gamma$=7.0. The value of 7.0 shows decreased diversity among generated objects, and the value of 3.0 shows decreased quality of generated objects, both resulting in lower performance. The classifier guidance's default value of 5.0 was also used in previous works [1, 2], and it was found to be effective also in our work.
>
> Table R1: Ablation on the classifier guidance using LVIS on ResNet-50.
> | **Classifier Guidance** ($\gamma$) |   $\mathbf{AP^{box}}$  | $\mathbf{AP^{mask}}$  |  $\mathbf{AP^{box}_r}$ | $\mathbf{AP^{mask}_r}$ |
> |------------|:----------------:|:---------------:|:----------:|:----------:|
> | 3.0     |   35.21        |   31.35     |   26.59    | 24.33
> | **5.0 (default)** |     **35.56** |    **31.81**    |    **28.14**   | **25.93**|
> | 7.0     |     35.32        |   31.58     |   26.89    | 24.68 |
>
> `W2. Ablation on reduced synthetic examples for competing methods.`
>
> In Tables R2 and R3, we borrow the experimental results based on a reduced quantity of synthetic examples from the original papers (BSGAL [2] and DiverGen [3]). **Previous works require more synthetic examples to saturate** since their generation method is class-balanced, resulting in redundant synthetic objects (i.e., generation of similar objects and/or generation on already high-performing classes). Thus, we confirm that **the competing methods lack sample efficiency** as opposed to our MRCA framework.
>
> Table R2: Result on a reduced quantity of synthetic examples for BSGAL [2] using LVIS on ResNet-50.
> | **Method (# Gen)** |   $\mathbf{AP^{box}}$  | $\mathbf{AP^{mask}}$  |  $\mathbf{AP^{box}_r}$ | $\mathbf{AP^{mask}_r}$ |
> |------------|:----------------:|:---------------:|:----------:|:----------:|
> | BSGAL(12k)     |   34.90        |   -     |   -    | -
> | BSGAL(480k)     |   35.12        |   -     |   -    | -
> | BSGAL(1200k)     |   35.40        |   31.56     |   27.95    | 25.43
> | **MRCA (72k)**     |     **35.56** |    **31.81**    |    **28.14**   | **25.93**|
>
> Table R3: Result on a reduced quantity of synthetic examples for DiverGen [3] using LVIS on Swin-L.
> | **Method (# Gen)** |   $\mathbf{AP^{box}}$  | $\mathbf{AP^{mask}}$  |  $\mathbf{AP^{box}_r}$ | $\mathbf{AP^{mask}_r}$ |
> |------------|:----------------:|:---------------:|:----------:|:----------:|
> | DiverGen(300k)      |   49.65 | 44.01 | 45.68 | 41.11
> | DiverGen(600k)     |   50.67  | 44.99  | 48.52  | 43.63
> | DiverGen(1200k)     |   51.24        |   45.48     |   50.07    | 45.85
> | **MRCA (72k)**     |    **51.80**|   **45.91**|   **51.58**| **46.84**
>
> `W3. Peformance improvement seems task specific and findings (on hyperparameters) maybe not generalizable.`
>
> While we focus on the instance segmentation task in this paper, we believe that the idea of collaborative augmentation between the generator and the target task model can be further generalized to other tasks with synthetic data generation. For example, the image classification task can benefit from the notion of the collaborative augmentation.
>
> We admit that some hyperparameters may not be generalizable, but the classifier guidance (refer to W1) and the guidance scale (Table 7) seem to be generalizable under the same generation model. Also, we would like to clarify that the necessity of hyperparameter tuning is not a specific issue unique to our work, but rather a common aspect of most machine learning methods.
>
> `Q1. Clarification on whether the generated images are processed with different(evolved) segmentation model over rounds.`
>
> We first clarify that there are two segmentation models used. (1) $M_{\phi_r}$: the target instance segmentation model with parameters $\phi_r$ with respect to a round $r$, which is trained for the downstream task; (2) $\mathcal{S}$: a frozen dichotomous segmentation model (BiRefNet [4]) that extracts an object from a generated image. While the model $M_{\phi_r}$ evolves over rounds, the dichotomous segmentation model $\mathcal{S}$ is fixed across all rounds. We will revise the notation in Section 3.1 to explicitly denote the round dependency of $M_{\phi_r}$.
>
> `Q2. Clarification on object edges for the Map-Paste() function.`
>
> We did not use any methods on object edges (e.g., Gaussian blurring and alpha/Poisson blending) when pasting objects. Based on previous work [5] and our own evaluation, the edge handling methods did not have a meaningful effect on the overall performance.
>
> `Q3. More detail on the concept of information and how increased dataset information implies improved augmentation is required.`
>
> We appreciate the feedback on clarifying the concept of _information_ in Lemma 3.2 and its role in guiding better augmentation.
>
> `Clarification on the Information Metric:` The information metric $\mathcal{I_D}(M_{\phi})$ is a flexible measure that quantifies how effectively the model $M_{\phi}$ captures the characteristics of the dataset $\mathcal{D}$. It can be instantiated using simple metrics such as average accuracy or defined more formally, for example, as the reciprocal of the KL divergence between the data distribution $p_{\mathcal{D}}$ and the model's distribution $p_{M_{\phi}}$: $\mathcal{I_D}(M_{\phi})= {1\over{D_{KL}(p_{\mathcal{D}} || p_{M_{\phi}})}}$.
>
> `Implication on Better Augmentation:` A higher $\mathcal{I_D}(M_{\phi})$ means that the model better represents the data distribution, enabling more meaningful feedback $f_\phi$ to improve augmentation quality. For example, this feedback can balance class distributions by generating more objects for rare classes or focus on difficult examples to enhance learning.
>
> `Q4. "as round pass" → "as rounds pass".`
>
> We will fix this typo.
>
> `Q5. Clarification on how entropy is measured for data instances.`
>
> Since an instance segmentation model predicts a class label for each segment, we compute the entropy based on the classification logit given an input object image. Here, the whole image is set as the object proposal since we generate only a single object per image. Specifically, for the $i$-th object at round $r+1$, denoted as $o_{i,r+1}$, the entropy is calculated as $\text{entropy}(o_{i,r+1}; M_{\phi}) = -\sum_{j} p_j \log(p_j)$, where $p = c_{\phi}(\mathcal{F}(o_{i,r+1}))$ is the softmax-normalized class probability vector obtained from the classifier head $c_{\phi}$, given features $\mathcal{F}(o_{i,r+1})$ extracted by the backbone of the model $M_{\phi}$. Using this formulation, the entropy results in Section 4.4 are obtained as follows.
> * Figure 4.4(b) presents $\text{entropy}(o_{i,r+1}; M_{\phi_{r}})$, where the feedback model from the previous round ($M_{\phi_r}$) is used.
> * Figure 4.4(c), on the other hand, shows $\text{entropy}(o_{i,r+1}; M_{\phi_{0}})$, where the warm-up model ($M_{\phi_0}$) is applied.
>
> `Q6. Why use a weaker ResNet backbone when Swin-L significantly outperforms it?`
>
> Since the overall performance trend remained consistent across different backbone models, we chose a lighter model **to enable more extensive ablation studies** within a given time. Notably, the performance improvement reported in Tables 1 and 2 is even larger when using the Swin-L backbone, supporting that our ResNet-based ablation results do not overstate the effectiveness of MRCA.
>
> The ResNet backbone is more memory-efficient and can be trained on a 24GB RTX 3090, whereas the Swin-L backbone requires a higher-memory setup, such as a 48GB A6000 or larger. We believe that both models are important, as they represent a trade-off between model size and performance, which is an important consideration for practical deployment, particularly in on-device scenarios.
>
> `Q7. Can augmented objects be overlapped?`
>
> Yes, the objects can be overlapped. When an object is overlapped, its object mask is adjusted to eliminate the obscured portion. This is a common practice in the relevant studies [1, 2, 3].
>
> ---
> Overall, we will incorporate these responses and additional results into the final version of the paper. We hope that most of your concerns have been addressed and would be glad to engage in further discussion if needed.
>
> ---
> [1] "X-paste: Revisiting scalable copy-paste for instance segmentation using clip and stablediffusion", In ICML, 2023
>
> [2] "Generative active learning for long-tailed instance segmentation", In ICML, 2024
>
> [3] "Diversify your vision datasets with automatic diffusion-based augmentation", In CVPR, 2024
>
> [4] "Bilateral reference for high-resolution dichotomous image segmentation", In CAAI AIR, 2024
>
> [5] "Simple Copy-Paste is a Strong Data Augmentation Method for Instance Segmentation", In CVPR, 2021

---

> > ### Comment · Reviewer_CkKJ · 2025-08-04
> >
> > We thank the authors for their clarifications, and retain the original rating as appropriate.

---

> ### Author Response · Authors · 2025-08-04
>
> We sincerely appreciate the reviewer's response and thoughtful feedback. We are glad to see that all questions have been thoroughly clarified and appreciate your decision to keep a positive score for our work.
>
> If there are any further questions, we would be more than pleased to address them.
>
> Warm regards,
>
> Authors

---

### Official Review · Reviewer_KqRM · 2025-07-03

**Clarity:** 2
**Significance:** 1
**Originality:** 2
**Rating:** 4
**Confidence:** 4

**Summary:**

This paper proposes a novel framework called Multi-Round Collaborative Augmentation (MRCA) to tackle the challenges of class imbalance and annotation costs in long-tail instance segmentation. MRCA operates by forming a feedback loop between a diffusion-based generative model and an instance segmentation network, selectively generating additional synthetic objects for classes that are either underrepresented or challenging for the model. By iteratively incorporating these synthetic samples across multiple rounds, MRCA progressively enhances the training dataset. In each round, both the number and size of generated objects are adaptively adjusted according to the current performance of the segmentation model, aiming to maximize sample efficiency. On benchmarks such as LVIS, MRCA achieves higher performance than existing state-of-the-art methods using only 6% as much synthetic data, and ablation studies demonstrate the contribution of each core component.

**Questions:**

1) In cases where the instance segmentation model trains much faster than object generation, have the authors observed any decline in the effectiveness of feedback-driven augmentation when the round gap is increased?
2) For rare classes that are inherently small in real images, have the authors experienced any negative effects on detection performance from scaling these objects up during training? I’m also curious if the authors have considered or tried any alternative resizing strategies for these cases.
3) While MRCA demonstrates strong performance with only 6% of the synthetic data compared to previous methods, I am curious whether further improvements are possible if the generation budget is increased to match that of methods like BSGAL or X-Paste+CLIP. (~1200k of #Gen Objects) Have the authors explored the performance of MRCA under higher generation budgets?

**Ethical Concerns:**

["NO or VERY MINOR ethics concerns only"]

**Final Justification:**

The authors have thoroughly addressed my concerns in the rebuttal, including issues related to feedback timing, object resizing, and sample efficiency. Given the clarity of the responses and the novelty of the proposed feedback-guided generation framework, I have updated my rating to borderline accept.

**Limitations:**

The accuracy-based resizing strategy involves artificially enlarging rare or low-accuracy objects rather than preserving their natural size distribution. This may disturb the true data size distribution, which could be particularly problematic in domains like medical imaging, where the physical size of objects often carries important diagnostic meaning. In such cases, this approach might introduce bias or lead to unintended side effects in model training and deployment.

**Quality:**

3

**Strengths And Weaknesses:**

Strengths:
- Among prior works addressing long-tailed instance segmentation with generative models, directly leveraging feedback from the instance segmentation model to guide the generative process is a fresh and interesting idea. The proposed framework is novel and, to my knowledge, has not been seen in previous literature.
- The paper conducts thorough ablation studies and analyses for each key component (such as class-wise budget allocation, model gradient feedback, and accuracy-based resizing), which provides valuable insights into the contribution of each part.
- The authors also provide reproducible code, which is a strong point in terms of transparency and research usability.

Weaknesses:
- Depending on the relative speeds of instance segmentation training and object generation, the required round gap in Equation 6 may need to be increased. For example, if the segmentation model is lightweight and trains much faster than object generation, a larger gap (e.g., 3 or more rounds) might be needed to prevent the training process from stalling. In this situation, the feedback used to guide object generation would come from an older model rather than the most recent checkpoint. This could reduce the method’s ability to adapt quickly to the model’s current weaknesses, and might weaken the core advantage of feedback-driven augmentation, especially in fast training scenarios.
- Limitation in accuracy-based resizing strategy is that some rare classes are inherently small in real life (e.g., coin, ant..) Enlarging these objects during training may harm the model’s ability to recognize them in realistic contexts where their small size is important.
- While the paper shows impressive sample efficiency given a fixed generative budget (e.g., 6%), I wonder whether constraints on synthetic data generation are as significant in practiceas reducing human annotation costs. The importance of being sample-efficient with generative models may not be as high as with real human-labeled data, so I am somewhat uncertain about the broader impact or significance of the proposed efficiency gains in this context.

---

> ### Author Rebuttal · Authors · 2025-07-30
>
> We sincerely appreciate the reviewer's insightful comments, which have significantly contributed to improving the quality of our paper. Below, we address each of the concerns in detail.
>
> `Q1(W1). In the case where the instance segmentation model trains much faster than object generation, is there any decline in the effectiveness of feedback-driven augmentation when the round gap is increased?`
>
> We first clarify that, when instance segmentation training is much faster than object generation, MRCA waits for the generation of objects from the **previous** round rather than increasing the round gap. **This stall is minimized** by adjusting two experimental settings. (1) **The number of GPUs assigned for each task** (instance segmentation training and object generation) is chosen to keep the speed similar between the two tasks. (2) **The number of object generations (72k)** is further set to better match the speed of training.
>
> We assigned 4 GPUs and 3 GPUs for instance segmentation training and object generation, respectively. For a lightweight segmentation model, for example, we can assign 2 GPUs and 5 GPUs instead. Also, generating 36k objects instead of 72k is a reasonable option according to Table 8.
>
> In summary, by adjusting the two parameters of MRCA, we exploit the **up-to-date** feedback from the previous round while minimizing the stall between instance segmentation training and object generation.
>
> `Q2(W2). Can scaling rare classes lead to negative effects? Are there any alternative approaches tested for resizing?`
>
> This is a very good point. Especially in domains like medical imaging, as you mentioned, where the scale of the images is usually consistent, the size of the objects conveys important contextual information, and altering it may distort the natural size distribution. When using MRCA on such data domains, we agree that using resizing strategies should be prohibited or be applied only on a small scale. This domain-specific issue will be included in Appendix D. Limitations.
>
> However, in the context of ***real-world images* on which we test (LVIS, PASCAL, OpenImages)**, the size of the objects varies significantly depending on the resolution or scale of the image, even within the same class. Please consider a person depicted in a close-up photograph and in a landscape photograph. Also, for rare classes, because the number of objects is small (0--9), their size distribution does not carry statistically significant meaning.
>
> As a result, we haven't observed any critical negative effects on detection performance. By setting the maximum scaling coefficient ($s_{max}=4.0$ in Algorithm 4) properly, we prevent objects from occupying the entire image or high-performing objects from becoming too small. Furthermore, it keeps the objects from deviating largely from their natural sizes. More importantly, as the performance of rare objects is improved in the upcoming rounds, their scale will approach their natural sizes.
>
> `Q3. Have the authors explored the performance of MRCA under higher generation budgets (i.e., 72k → 1200k)? (W3) Also, are constraints on synthetic data generation as significant in reducing human annotation costs?`
>
> `Higher Generation Budget:` Great point, as we also wanted to verify in such a case where the number of generated objects is similar to that of previous works. In **Table 8**, we tested 144k generated objects but observed only small improvements compared to 72k. Because of such observations of early saturation and **long training/generation time**, we have decided not to increase the generation budget over 144k.
>
> We believe that since a lot of redundant generations from previous works [1,2,3] (i.e., generation of similar objects and/or generation on already high-performing classes) are removed by our pipeline, MRCA can outperform previous works with only a small number of object generations. Using multiple generation models could be a way to further improve our pipeline, which we leave as future work.
>
> `Sample Efficiency with Synthetic Data:` The significance of sample efficiency is indeed larger in real human-labeled data compared to synthetic data, as you pointed out. However, we contend that **running GPUs for synthetic data generation is also costly**. For example, running eight RTX 3090 GPUs for 3 weeks to generate and use 1200k objects incurs a cost of **766.08 US\$** (\$0.19/hr in vast.ai), whereas our MRCA framework requires only **51.68 US\$**. For the broader impact, using fewer GPUs reduces electricity consumption and helps protect the environment.
>
> ---
> Overall, we will incorporate these responses and additional results into the final version of the paper. We hope that most of your concerns have been addressed and would be glad to engage in further discussion if needed.
>
> ---
> [1] "X-paste: Revisiting scalable copy-paste for instance segmentation using clip and stablediffusion", In ICML, 2023
>
> [2] "Generative active learning for long-tailed instance segmentation", In ICML, 2024
>
> [3] "Diversify your vision datasets with automatic diffusion-based augmentation", In CVPR, 2024

---

> > ### Comment · Reviewer_KqRM · 2025-08-05
> >
> > The authors have addressed all of my concerns in the rebuttal. I believe this is a paper with an interesting idea, and accordingly, I am updating my rating to borderline accept. Thank you for the effort put into the detailed rebuttal.

---

> ### Author Response · Authors · 2025-08-05
>
> We sincerely appreciate the reviewer's thoughtful reassessment and encouraging feedback. We're glad our clarifications addressed all your concerns, and we appreciate your recognition of our core idea of multi-round collaborative augmentation. We will incorporate all of your feedback into the final version of our paper.
>
> If there are further questions, we will be delighted to answer them anytime.
>
> Sincerely,
>
> Authors

---

### Note · Authors · 2025-08-13

Dear Reviewers, ACs, SACs, and PCs:

We thank the reviewers for recognizing **the novelty of our multi-round collaborative augmentation (MRCA)** (KqRM, CkKJ, SKJ8, 3V2k), **the logical structure of our presentation and illustration** (SKJ8, 3V2k), and **the extensive evaluation** (KqRM, CkKJ, SKJ8, 3V2k). Our paper received favorable initial ratings of 4, 4, 4, and 3, which increased to **4**, **4**, **4**, and **4**. This improvement clearly indicates the success of our rebuttal in addressing almost all concerns and clarifying the novelty, necessity, and empirical strength of our MRCA framework.

Our responses to the main concerns are summarized as follows.

1. Importance of Sample Efficiency in Synthetic Data

- While acquiring real human-labeled data is costly, we have emphasized that generating synthetic data also entails substantial GPU costs. MRCA achieves superior performance with only 6% of the generation cost required by state-of-the-art methods. $\rightarrow$ Reviewer KqRM is satisfied with our response.

2. Further Ablation Studies and Empirical Results

- We have provided extensive ablation studies on classifier guidance, resizing, and fine-grained class budget optimization, as well as more results for reduced-example scenarios across competing baselines. $\rightarrow$ Reviewer CkKJ is satisfied with our response.

3. Theoretical Foundations for Feedback Benefits

- We have added a rigorous analysis that defines information gain and formalizes the conditions under which model feedback enhances augmentation quality. $\rightarrow$ While Reviewer CkKJ is satisfied with our response, Reviewer 3V2k is *partially* satisfied. Still, our assumption aligns with widely-adopted practices in recent data augmentation literature.

4. Clarifications on Methods and Experiment Settings

- We have improved the clarity of the paper by elaborating on our framework and by providing additional details on the experiment settings and results. $\rightarrow$ Reviewers KqRM, CkKJ, and 3V2k are satisfied with our response.

Meanwhile, although Reviewer SKJ8 did not explicitly acknowledge our rebuttal, we are confident that our responses have fully addressed all points raised.

`Conclusion`: **With all four reviewers supporting acceptance**, we hope that MRCA's practical efficiency, solid theoretical basis, and compelling empirical results will lead to a positive final decision and a valuable contribution to the NeurIPS community.

---

### Decision · Program_Chairs · 2025-09-17

**Decision:**

Accept (poster)

**Comment:**

***(a) Scientific Claims and Findings***

The paper proposes a novel framework called Multi-Round Collaborative Augmentation (MRCA) for long-tail instance segmentation. The core idea is to create a feedback loop between a diffusion-based generative model and an instance segmentation network. The generative model selectively produces synthetic objects for classes that are either underrepresented or difficult for the current model to segment. This process is performed over multiple rounds, with the number and size of synthesized objects for each class dynamically adjusted based on the model's state. The authors claim that MRCA enhances sample efficiency and outperforms state-of-the-art methods while using only 6% of the data generation required by competing methods.

***(b) Strengths***
+ Novelty: The framework is considered a fresh and interesting idea, leveraging feedback from an instance segmentation model to guide the generative process. This is a new approach to generative data augmentation.
+ Empirical Performance: The paper provides compelling experimental evidence that MRCA outperforms state-of-the-art methods while using significantly less synthetic data. The authors conducted extensive evaluations, which reviewers found to be thorough.
+ Methodological Soundness: The framework is conceptually strong and intuitive. The logical structure and clear presentation of the method were also praised by reviewers.

***(c) Weaknesses***
+ Theoretical Superficiality: The theoretical analysis was initially criticized as being "superficial" and only "marginally insightful". The paper's core assumption that feedback improves augmentation quality lacked formal conditions.
+ Generalizability Concerns: A reviewer noted that the performance improvements might be task-specific and that the findings on optimal hyperparameters may not be fully generalizable. They also suggested the method's impact might be diminished when using stronger backbone models, though the authors provided a counterargument.
+ Practical Limitations: One reviewer raised concerns about the resizing strategy, noting that it might be problematic for classes that are inherently small and could disturb the true data size distribution, particularly in domains like medical imaging.


***(d) Reasons for Decision to Accept***

I recommend Acceptance as poster for this paper. The core idea is highly innovative and addresses a critical problem in the field: the inefficiency of data generation for long-tail instance segmentation. The authors' approach of using a feedback loop to guide a diffusion model is a significant step forward.

***(e) Summary of Rebuttal and Discussion***

While the initial submission had some theoretical and clarity issues, the authors provided a detailed and convincing rebuttal. They successfully addressed all major concerns, providing additional ablation studies, clarifying their experimental setup, and adding a more rigorous analysis of their theoretical foundations. The authors received positive initial reviews but were asked to address several key points. They provided a new ablation study on classifier guidance, clarified their experimental setup, and added a more formal theoretical analysis to address the "superficial" critique. They also provided convincing arguments and additional data to address concerns about sample efficiency, object resizing, and the generalizability of their findings. The rebuttal successfully addressed all major concerns raised by the reviewers, leading to a unanimous favorable rating among all four. This clear consensus among the reviewers, combined with the paper's strong empirical results and practical efficiency, justifies its acceptance as a valuable contribution to the NeurIPS community.